



# A Local Analytical Optimal Nudging for assimilating AMSR2 sea ice concentration in a high-resolution pan-Arctic coupled ocean (HYCOM 2.2.98) and sea ice (CICE 5.1.2) model

Keguang Wang[1,*], Alfatih Ali[2], and Caixin Wang[1]

[1]Division of Ocean and Ice, Norwegian Meteorological Institute, Oslo, Norway
[2]Division of Ocean and Ice, Norwegian Meteorological Institute, Bergen, Norway
[*]**Correspondence:** Keguang Wang (keguang.wang@met.no)

**Abstract.** A Local Analytical Optimal Nudging (LAON) is introduced and thoroughly evaluated for assimilating the Advanced Microwave Scanning Radiometer 2 (AMSR2) sea ice concentration (SIC) in the Norwegian High-resolution pan-Arctic ocean and sea ice Prediction System (NorHAPS). NorHAPS is a developing high-resolution ($3 - 5$ km) pan-Arctic coupled ocean and sea ice modeling and prediction system based on the HYbrid Coordinate Ocean Model (HYCOM Version 2.2.98) and the Los

Alamos multi-category sea ice model (CICE Version 5.1.2), with the LAON for data assimilation. In this study, our focus is on the LAON assimilation of AMSR2 SIC, which is designed to update the model SIC in every time step such that the analysis will eventually reach the optimal estimate. The SIC innovation (model minus observation) is designed to be proportionally distributed to the multiple sea ice categories.

A twin experiment is performed with and without the LAON assimilation for the period 1 January 2021 to 30 April 2022.

The results show that the LAON assimilation greatly improves the simulated sea ice concentration, extent, area, thickness and volume, as well as the sea surface temperature (SST). It also produces significantly more accurate sea ice edge and marginal zone (MIZ) than the observed AMSR2 SIC that is assimilated when evaluated against the Norwegian Ice Service (NIS) ice chart. The results are also compared with the Copernicus Marine Environment Monitoring Service (CMEMS) operational SIC analyses from NEMO, TOPAZ4 and neXtSIM which use ensemble Kalman filters and direct insertion for data assimilation.

It is shown that the LAON assimilation produces significantly lower integrated ice edge error (IIEE) and integrated MIZ error (IME) than the CMEMS SIC analyses when evaluated against the NIS ice chart. The LAON also produces a continuous and smooth evolution of sub-daily SIC, which avoids abrupt jumps often seen in other assimilated products. This efficient and accurate method is promising for data assimilation in global and high-resolution models.

## 1   Introduction

Arctic sea ice is one of the most sensitive components in Earth's climate system. In recent decades it has been undergoing a dramatic change, where vast areas previously covered by multi-year sea ice are now dominated by younger, thinner ice or even seasonally ice-free (Comiso, 2012; Meier et al., 2014; Kwok, 2022; Stroeve and Notz, 2018; Kacimi and Kwok, 2022; Constable et al., 2022). While this change is opening new opportunities for accessing the Arctic (Smith and Stepheson, 2013;



PAME, 2020; Berkman et al., 2022; Constable et al., 2022), it also brings higher environmental risks and climate challenges to

the Arctic (Emmerson and Lahn, 2012; Meier et al., 2014; Dammann et al., 2018; Cohen et al., 2020). To effectively manage the opportunities and risks, sound measures are urgently needed to ensure adequate sustainable development, safe operation and ecosystem-based management. Accurate and timely sea ice forecast is thus becoming more and more important for the planning and regulation of the activities in the Arctic (Eicken, 2013; Jung et al., 2016).

Accurate sea ice forecast depends strongly on the initial conditions, which are commonly prepared by data assimilation via

a combination of model simulations and observations. A variety of sea ice data assimilation methods have been developed in the last two decades, including direct insertion (Caya et al., 2010; Posey et al., 2015; Williams et al., 2021), nudging (Lindsay and Zhang, 2006; Caya et al., 2010; Wang et al., 2013; Tietsche et al., 2013; Fritzner et al., 2018), optimal interpolation (OI; Zhang et al., 2003; Stark et al., 2008; Wang et al., 2013), three-dimensional variational (3DVar; Caya et al., 2010; Buehner et al., 2013; Blockley et al., 2014; Waters et al., 2015), and ensemble Kalman filter (EnKF; Lisæter et al., 2003; Sakov et al.,

2012; Mathiot et al., 2012; Yang et al., 2014; Fritzner et al., 2018). It is noteworthy that when assimilating the same sea ice observations, the forecast quality tends to be quite similar regardless of the different assimilation methods (e.g. Caya et al., 2010; Fritzner et al., 2018).

In recent years, with the continuous development of coupled atmosphere, ocean and sea ice models and increasing of model spatial resolutions, there is a growing interest in computationally efficient data assimilation methods for providing accurate and

timely high-resolution forecasts. In the present study, a Local Analytical Optimal Nudging (LAON) is introduced to provide an efficient and accurate method to assimilate the high-resolution (3.125 km) Advanced Microwave Scanning Radiometer 2 (AMSR2) sea ice concentration (SIC) into the multi-category Los Alamos sea ice model CICE (Hunke et al., 2015) in the coupled ocean and sea ice model HYCOM-CICE. The extra computational cost for the LAON assimilation is negligibly small, being about 5% of the free run in the present study.

LAON is a further development of the Combined Optimal Interpolation and Nudging (COIN; Wang et al., 2013). The originally empirical treatments of the combination of OI and nudging in COIN has been upgraded as a theoretically self-contained optimization (see section 2). The analysis in LAON is designed to be gradually nudged to the optimal estimate, rather than nudged towards the observation in the ordinary nudging (Anthes, 1974). The LAON assimilation is only performed forward in time, with the optimal nudging coefficient deduced analytically. This is different from the variational optimal

nudging (Zou et al., 1992; Vidard et al., 2003), where the optimal nudging coefficient is obtained through parameter estimation with a complex minimization procedure using integrations of both direct and adjoint models.

The present study is organized as follows. Section 2 introduces the coupled HYCOM-CICE model system, together with the LAON that is coded in the multi-category CICE model for SIC data assimilation. In section 3, a variety of observation data are introduced for model evaluation, including SIC, sea ice thickness (SIT), sea surface temperature (SST) and sea surface salinity

(SSS), together with three Copernicus Marine Environment Monitoring Service (CMEMS) SIC analyses (NEMO, TOPAZ4 and neXtSIM). In section 4 we perform a twin experiment with and without the LAON assimilation, and evaluate the effect of the LAON SIC assimilation on the modeled ocean and sea ice variables. In section 5 we further compare the LAON simulation with the CMEMS SIC analyses from the NEMO, TOPAZ4 and neXtSIM models, and evaluate their skills in simulating the





sea ice edge and marginal ice zone (MIZ) against the Norwegian Ice Service (NIS) ice charts. In section 6, we discuss some
general issues relating to the data assimilation and model evaluation. The conclusions are given in section 7.

## 2   Model and data assimilation

NorHAPS is a developing high-resolution pan-Arctic ocean and sea ice modeling and prediction system running at the Nor-
wegian Meteorological Institute. It is based on the coupled HYCOM-CICE model developed at the Nansen Environment and
Remote Sensing Center (NERSC; https://github.com/nansencenter/NERSC-HYCOM-CICE), with the LAON for data assimi-
lation (Wang and Ali, 2023). Its free run version (no data assimilation) is currently run operationally and delivers daily 10-day
forecasts of sea surface height and sea surface velocity components to CMEMS as snapshots at 15-minutes frequency (Ali
et al., 2021), which is available for free download from CMEMS (https://doi.org/10.48670/moi-00005).

### 2.1   HYCOM-CICE model

The coupled HYCOM-CICE model is based on the HYbrid Coordinate Ocean Model (HYCOM version 2.2.98; Bleck, 2002)
and the Los Alamos Community Ice Code (CICE version 5.1.2; Hunke et al., 2015). Their coupling is accomplished using the
Earth System Modeling Framework (ESMF, version 8.0). The model domain covers the Arctic and North Atlantic, with the
horizontal grid length varying from 3.2 to 5.1 km and a total grid number of $1600 \times 1520$ (Fig. 1).

The HYCOM model uses a 50-layer hybrid z-isopycnal vertical coordinate, with the isopycnal coordinate in the stratified
ocean and the z-coordinate in the unstratified surface mixed layer. The top 10 layers are chosen to be fixed z-layers, with
the thickness of the top layer being 1 m. This allows for better resolved upper ocean processes. The bathmetry is from the
General Bathymetric Chart of the Ocean (GEBCO_2014, https://www.gebco.net/), with a 30 arc-second global grid. The 3D
non-tidal lateral boundary forcing is from the CMEMS global NEMO analysis (https://doi.org/10.48670/moi-00016), and the
tidal lateral boundary uses hourly tidal currents and elevation from the FES2014 global model (Lyard et al., 2021). The river
runoff is extracted from the Arctic-Hype Hydrographic model of SMHI (Lindström et al., 2010). The model barotropic and
baroclinic time steps are 7.5 seconds and 2.5 minutes, respectively. The atmospheric forcing is from the ECMWF IFS HRES
analysis, including 10 m wind velocity, 2 m air temperature and due temperature, mean sea level pressure, total cloudiness,
total precipitation, surface solar radiation downwards and surface net solar radiation, and surface net thermal radiation. These
atmospheric forcing is read in via HYCOM, and transferred to CICE through the ESMF. The surface fluxes are parmaterized
based on the COARE 3.0 bulk algorithm (Fairall et al., 2003).

Since the focus of the present study is on the LAON assimilation of SIC, more effort is made here to describe the CICE
model, in particular those related to the evolution of SIC. CICE is thus far one of the most widely used sea ice models for
climate studies, and is now becoming more and more involved in short-term and subseasonal-to-seasonal sea ice predictions.
It is a dynamic and thermodynamic, elastic-viscous-plastic (EVP), multiple ice-thickness category sea ice model (Hunke and
Dukowicz, 1997). It is a reformulation of the earlier viscous-plastic model of Hibler (1979, 1980), with an artificial elastic term
introduced to enhance the computational efficiency (Hunke and Dukowicz, 1997). Sea ice conditions in CICE are described by

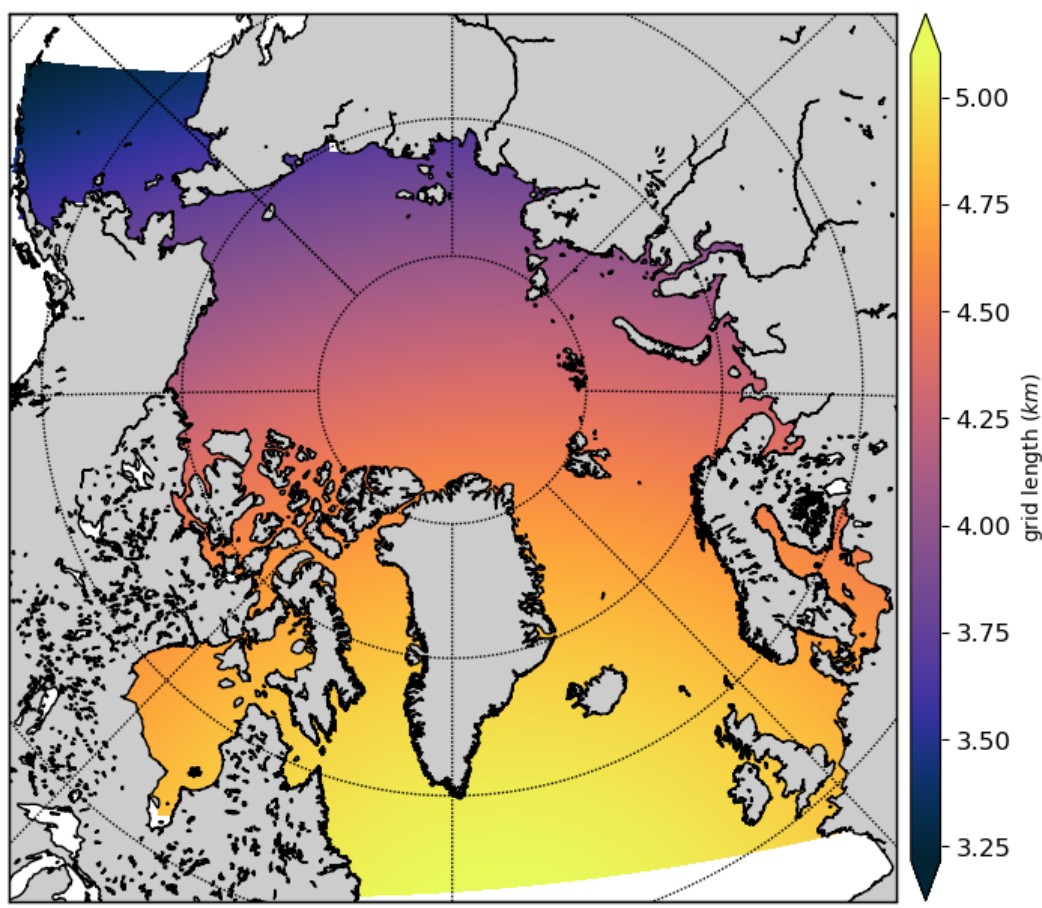

**Figure 1.** Model domain and grid length of the coupled HYCOM-CICE model.

the ice thickness distribution (ITD) function, $g(\mathbf{x}, h, t)$, determined by the following equation (Thorndike et al., 1975; Hibler, 1980; Hunke et al., 2015):

$$\frac{\partial g}{\partial t} = -\nabla \cdot (g\mathbf{u}) - \frac{\partial (fg)}{\partial h} + \psi \tag{1}$$

where $g(\mathbf{x}, h, t)dh$ is defined as the fractional area covered by ice in the thickness range $(h, h+dh)$ at a given time $t$ and location

$\mathbf{x} = (x, y)$. $\mathbf{u}$ is sea ice velocity, $f$ is the rate of thermodynamic ice growth, and $\psi$ is a ridging redistribution function. In CICE, Eq. (1) is solved by partitioning the ice cover in each grid cell into discrete thickness categories. For each category $n$ with a lower thickness bound $H_{n-1}$ and upper bound $H_n$, integrate Eq. (1) for $h$ we get (Thorndike et al., 1975),

$$\frac{\partial a_n}{\partial t} = -\nabla \cdot (a_n \mathbf{u}) - \frac{\partial (fa_n)}{\partial h} + \Psi \tag{2}$$





where $\Psi$ is the accumulative ice redistribution function, $a_n$ is the accumulative ITD function or ice fraction for the $n$th ice category, defined as the fractional area covered by ice in the thickness range $(H_{n-1}, H_n)$,

$$a_n = \int_{H_{n-1}}^{H_n} g(\mathbf{x}, h, t)dh \tag{3}$$

Equation (2) is solved by splitting it into three pieces, namely a horizontal two-dimensional transport, a vertical one-dimensional transport in thickness space, and a redistribution of the ice in the thickness space through a ridging model. In our simulations, the original five category ITD ($kcatbound = 0$) is selected to describe the ice conditions, and the vertical snow and ice are resolved with seven ice layers and one snow layer for each ice category.

The ice velocity $\mathbf{u}$ is calculated from the sea ice momentum equation that account for air and water drags, Coriolis force, sea surface tilt, and the divergence of internal ice stress. The evolution of internal stress is described by the EVP rheology (Hunke et al., 2015), with the ice strength reformulated according to Rothrock (1975), and the advection using the incremental remapping scheme (Lipscomb and Hunke, 2004). The subgrid sea ice deformation and the redistribution of various ice categories follow Rothrock (1975), with a modified expression for the participation function (Lipscomb et al., 2007). In this study, the revised EVP approach (Bouillon et al., 2013) is used to remove the artificial deformation features.

The sea ice thermodynamic growth/melting rate $f$ is determined by solving the one-dimensional vertical heat balance equations for each ice thickness category and snow, using the mushy-layer scheme that also accounts for the evolution of sea ice salinity (Turner et al., 2013). The upper snow/ice boundary is assumed to be balanced under shortwave and longwave radiations and sensible, latent and conductive heat fluxes when the surface temperature is below freezing. When the surface is warmed up to the melting temperature, it is held at the melting temperature and the extra heat is used to melt the snow/ice surface. The bottom sea ice boundary is assumed to be at dynamic balance, growing or melting due to the heat budget between ice conductive heat flux and the under-ice oceanic heat flux. The lateral melting is calculated by the default parameterization in CICE according to Maykut and Perovich (1987). The melt pond is assumed to occur only on level ice, following the LEVEL-ICE melt pond parameterization (Hunke et al., 2013).

## 2.2 LAON data assimilation

Nudging is an efficient four-dimensional data assimilation method (Stauffer and Seaman, 1990). It has been about an half century since the nudging method was first applied for data assimilation in geoscience (Anthes, 1974), in which the model values are designed to be gradually nudged towards the observation

$$\frac{\partial X}{\partial t} = F(X, t) + G[X_{obs} - O(X)] \tag{4}$$

where $X$ denotes any concerned variables to be assimilated, $F(X, t)$ denotes the nonlinear model processes, e.g. the processes shown by the right side of Eq. (2), $X_{obs}$ is the corresponding observations, $O$ is observation operator, and $G$ is the nudging matrix.

In contrast to the ordinary nudging, the LAON nudges the model results to the optimal estimate. Here we provide a detailed deduction of the theoretical framework for the LAON assimilation, and then describe the special treatment for the multi-





category sea ice situations. Following Wang et al. (2013), we only consider the local variance of the model and observations, i.e. the spatial covariance between different grids is assumed null in the model (so far, uncertainties of sea ice observations only contain local variance or standard deviation). Such a treatment can significantly simplify the coding and reduce the computation cost.

The LAON assimilation is coded in the physical model (here CICE), and performed online with the physical model in every time step, thus producing an overall continuous evolution of the assimilated fields. This is different from other nudging method used for sea ice data assimilation, which were performed offline and applied once a day (e.g. Lindsay and Zhang, 2006; Tietsche et al., 2013) or every 10 days (e.g. Fritzner et al., 2018). Such a high-frequency assimilation effectively avoid model instabilities due to large changes during the assimilation often occurred in the previous studies (e.g. Lindsay and Zhang, 2006;

Mathiot et al., 2012; Fritzner et al., 2018). As a result, no particular post processing is applied after data assimilation.

Figure 2 shows a schematic picture illustrating the assimilation procedures using OI, 3DVar, LAON and EnKF. In general, the OI and 3DVar are equivalent when the model and observation error covariances are the same (Lorenc, 1986). As shown in Fig. 2, we denote $X$ as any concerned variables to be assimilated. $X_k^{obs}$ is the $k$th observation at time $T_k$, $X_k^-$ and $X_k^+$ are the model results before and after the assimilation at time $T_k$ when using OI or 3DVar for assimilation. The time period between

two successive observations, $\Delta T = T_k - T_{k-1} = N\Delta t$, where $\Delta t$ is the model time step. In the present study, the observation time step $\Delta T = 1$ day and the model time step $\Delta t = 2.5$ minutes, hence $N = 576$.

As a reference, the integration and assimilation using OI or 3DVar between two successive observations can be expressed as

$$X_k^- = X_{k-1}^+ + \int_{T_{k-1}}^{T_k} F(X,t)dt \tag{5}$$

$$X_k^+ = X_k^- + K(X_k^{obs} - X_k^-) = (1-K)X_k^- + KX_k^{obs} \tag{6}$$

where the integration of $F(X,t)$ denotes the model free run from $T_{k-1}$ to $T_k$, and $K$ is the Kalman gain which in the local situation as (e.g. Wang et al., 2013)

$$K = \frac{\sigma_{mod}^2}{\sigma_{mod}^2 + \sigma_{obs}^2} \tag{7}$$

where $\sigma_{mod}$ and $\sigma_{obs}$ are the standard deviations of the model and observations. Similarly, when using the EnKF, all the ensemble members will first be integrated after $N$ time steps from $T_{k-1}$, and then updated at time $T_k$, with the optimal estimate

being the assimilated ensemble mean which is close to the optimal estimate using OI or 3DVar.

In contrast to the OI, 3DVar and EnKF, the LAON assimilation is performed in every model time step (see Fig. 2), which can be expressed for the period from $T_{k-1}$ to $T_k$ as

$$\frac{\partial X}{\partial t} = \beta K(X_k^{obs} - X)$$

$$X(0) = X_k^- \tag{8}$$

where $\beta$ is a nudging coefficient, designed as a constant to be determined such that the overall LAON assimilation in the period

$[T_{k-1}, T_k]$ is equivalent to the single assimilation update using OI or 3DVar at time $T_k$ (see Fig. 2 and Eq. 6). The initial value



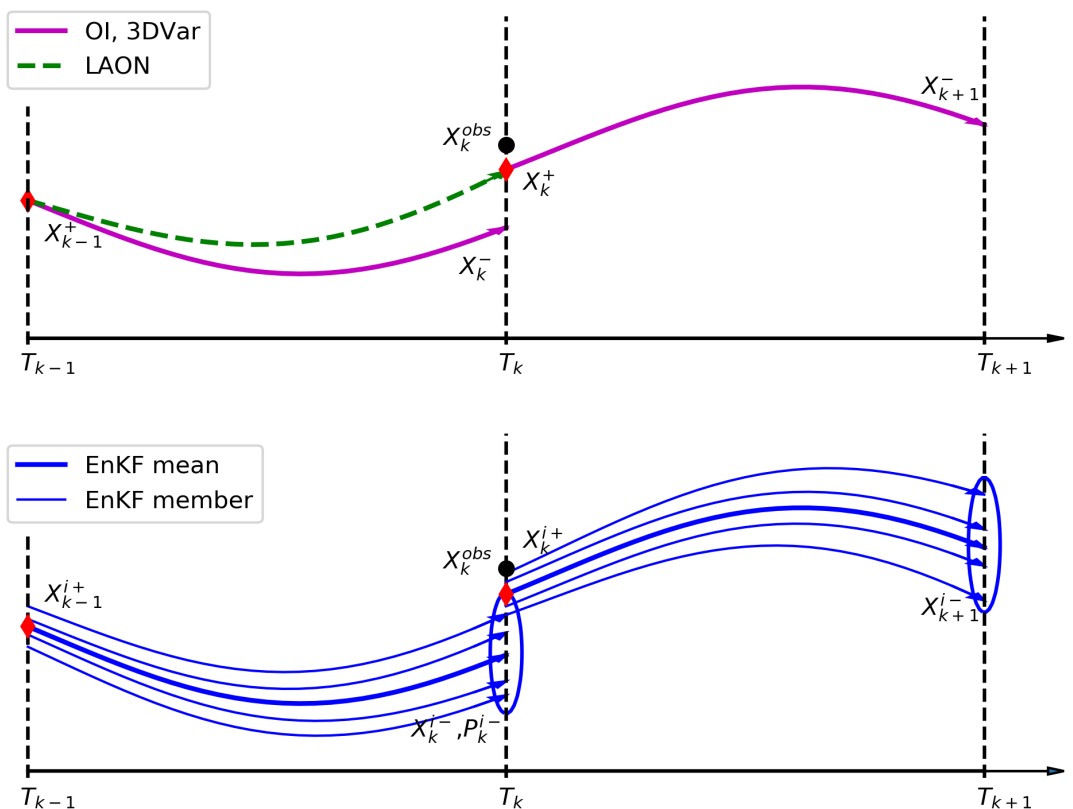

**Figure 2.** A schematic illustrating the assimilation procedures using OI, 3DVar, LAON (upper) and EnKF (lower). $X_k^{obs}$ (black dot) denotes the $k$th observation at time $T_k$; $X_k^-$ and $X_k^+$ (red diamond) denote the model results before and after assimilation at $T_k$. The superscript $i$ denotes the ensemble member in the EnKF. The observation time step is $\Delta T = T_k - T_{k-1} = N\Delta t$, with $\Delta t$ being the model time step. $\Delta T$ and $\Delta t$ are 1 day and 2.5 minutes in the present LAON Assimilation.

is from the integration of free run following Eq. (5). For simplicity, we denote the observation $X_k^{obs}$ as $X_{obs}$, denote $X_k^-$ as $X_0$, being the initial value when starting the LAON assimilation at $T_{k-1}$, and $X_k^+$ as $X_N$, being the final result at $T_k$ after the whole integration. The other intermediate results are denoted as $X_j$, where $j = 1, 2, \ldots, N-1$. Using a simple forward Euler scheme, the differential LAON assimilation equation (Eq. 8) can be discretized as

$$X_{j+1} = X_j + W(X_{obs} - X_j) = (1-W)X_j + WX_{obs} \tag{9}$$

where $W = \beta K \Delta t$ is the nudging weight. From Eq. (9) we get the contribution from the LAON assimilation at time step $j$ in terms of the initial value $X_0$ and the observation $X_{obs}$

$$X_j = (1-W)^j X_0 + [1 - (1-W)^j]X_{obs} \tag{10}$$

where $j = 1, 2, \ldots, N$. According to the original design for the LAON assimilation, Eq. (10) should be equivalent to the OI
or 3DVar single assimilation update in Eq. (6) when $j = N$. Noting here $X_0 = X_k^-$, we get the corresponding optimal nudging



weight

$$W = 1 - (1-K)^{\frac{1}{N}} \approx \frac{K}{N} \tag{11}$$

and the corresponding optimal nudging coefficient

$$\beta = \frac{W}{K\Delta t} \approx \frac{1}{N\Delta t} = \frac{1}{\Delta T} \tag{12}$$

which is approximate to the reciprocal of the observation time step. This indicates that the observation time step $\Delta T$ is exactly the optimal nudging time scale. For the model error standard deviation, following the earlier studies (Wang et al., 2013; Fritzner et al., 2018), we take

$$\sigma_{mod} = |X_{mod} - X_{obs}| \tag{13}$$

With the optimal nudging weight $W$ in Eq. (11), the LAON is ready for SIC assimilation in one-category sea ice models
using Eq. (9) such that

$$a_{ice,j+1} = a_{ice,j} + W(a_{obs} - a_{ice,j}) \tag{14}$$

where $a_{ice}$ and $a_{obs}$ are the total model SIC and observed SIC. For multi-category sea ice models, earlier studies designed their assimilations to modify the thinnest sea ice category (e.g. Lindsay and Zhang, 2006; Blockley et al., 2014). In the present multi-category CICE model, we apply a different formulation. When $a_{ice} > 0$, Eq. (14) can be rewritten as

$$a_{ice,j+1} = a_{ice,j}[1 + W(a_{obs}/a_{ice,j} - 1)] \tag{15}$$

Thus, a proportional formulation can be applied to update the ice and snow conditions for all the ice categories such that

$$a_{n,j+1} = a_{n,j}(1+\gamma) \tag{16}$$
$$v_{n,j+1} = v_{n,j}(1+\gamma) \tag{17}$$
$$v_{sn,j+1} = v_{sn,j}(1+\gamma) \tag{18}$$

where $v_n$ and $v_{sn}$ are ice and snow volumes for the $n$th ice category, and the rate of incremental innovation (see Eq. 15) is modified as

$$\gamma = W\left[\frac{a_{obs}}{max(a_{ice}, 0.1)} - 1\right] \tag{19}$$

where the function $max$ is used to avoid huge values when $a_{obs}/a_{ice} \gg 1$. In this proportional formulation (Eqs. 16 − 18), all the variables ($a_n, v_n$ and $v_{sn}$) are updated according to the same incremental innovation $\gamma$ (Eq. 19). Except for the situations
when the function $max$ is activated, this formulation keeps the actual SIT $h_n$ and snow depth $h_{sn}$ unchanged during the LAON assimilation. Similarly, the proportions of $a_n$, $v_n$ and $v_{sn}$ also remain unchanged when scaled with the total SIC, SIV and snow volume. This conservative property facilitates the maintenance of valid sea ice variables during the LAON assimilation.





For the situation when $a_{ice} = 0$ and $a_{obs} > 0$, we assume that new model ice will form with the actual SIT either being 0.5 m (Wang et al., 2013), or determined by an empirical formula (Fritzner et al., 2018)

$$h_{bar} = 0.02e^{2.8767a_{obs}} \tag{20}$$

In addition, we set the snow volume as 0.1 of the ice volume, sea ice salinity as 5 PSU, and sea ice temperature at the freezing temperature with the corresponding entropy. Equation (20) is used in the present study.

## 3  Data

We choose the following parameters to evaluate the effect of the LAON assimilation of SIC: SIC, sea ice extent (SIE), sea ice area (SIA), SIT, sea ice volume (SIV), SST, and SSS. These parameters are generally considered closely related to the SIC. In particular, SIE and SIA are direct products of SIC. As the main focus is on the assimilation of SIC, we here use three SIC observations and three CMEMS SIC analyses to thoroughly evaluate the LAON SIC assimilation. The SIT, SST and SSS data are mainly used to assess the effect of SIC assimilation on the model simulations. All the data were interpolated to the model grid using the nearest neighbour method.

### 3.1  SIC observations

In the present study, we use three SIC observation data, namely the AMSR2 SIC, the Special Sensor Microwave Imager/-Sounder (SSMIS) SIC and the NIS ice chart.

#### 3.1.1  AMSR2 SIC

The AMSR2 SIC data is from the University of Bremen (https://seaice.uni-bremen.de/data/amsr2/asi_daygrid_swath/n3125/ ,version5.4, accessed on 17.06.2022), being the latest version with the spatial resolution of 3.125 km (Melsheimer, 2019). The AMSR2 onboard the GCOM-W1 satellite is a remote sensing instrument for measuring weak microwave emission from the Earth, with a nominal incident angle of 55° and a swath width of 1450 km. The AMSR2 SIC here uses the same ARTIST sea ice (ASI) algorithm for the AMSR-E 89 GHz channel (Spreen et al., 2008), and is interpolated from the same swath data for the AMSR2 6.25 km SIC product to make the best use of the AMSR2's 89 GHz product (Gunnar Spreen, February 2022, personal communication). The AMSR2 SIC is the only data being assimilated in this study, and it is also used for evaluation in section 4. Figure 3 shows the SIC and its uncertainty (here standard deviation ) on 1 January 2021 from this product. The uncertainty is calculated following the same procedure in Spreen et al. (2008), where the overall error sums from three sources: the radiometric error from the bright temperature, the variability of the tie points, and the atmospheric opacity. It can be seen that the standard deviation is highest in the open water and lowest in the close drift ice, being about 0.25 when SIC = 0, and about 0.057 when SIC = 1 as in Spreen et al. (2008). The high uncertainty in the open water implies that the observed ice edge may be not very accurate. Its impact on data assimilation is further investigated in section 5 with evaluations against the NIS ice chart.





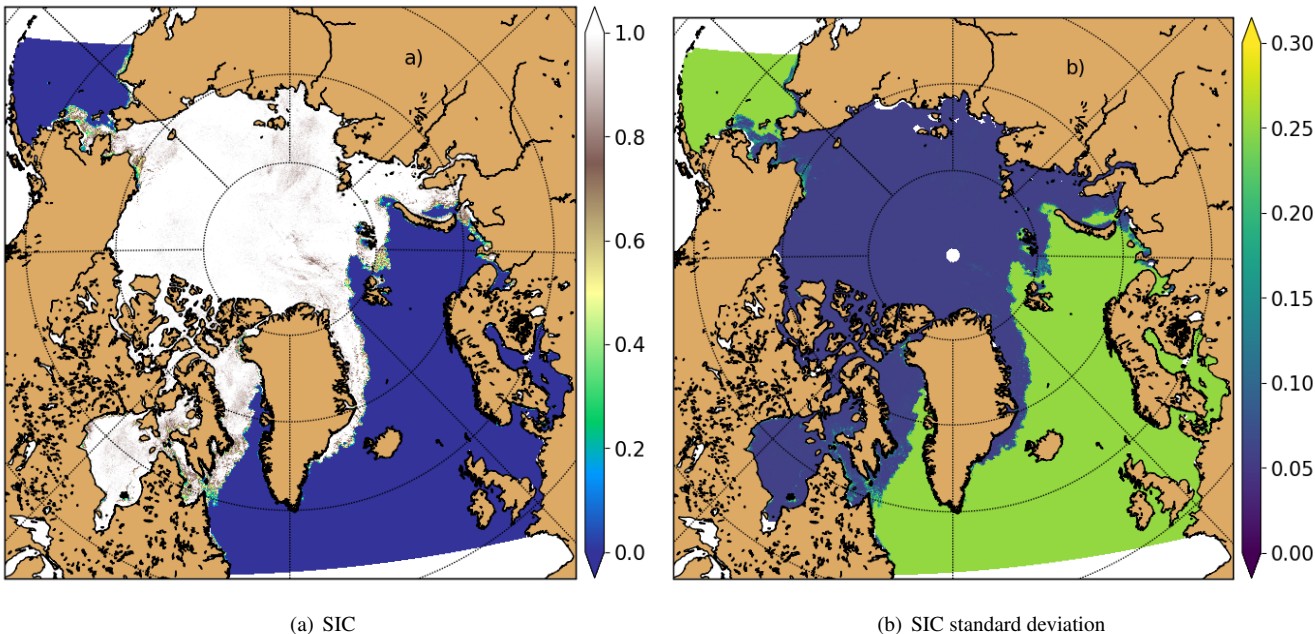

(a) SIC                                                    (b) SIC standard deviation

**Figure 3.** AMSR2 SIC and its uncertainty on 1 January 2021. The SIC data is obtained from the University of Bremen.

### 3.1.2  SSMIS SIC

The SSMIS SIC is from the EUMETSAT Ocean and Sea Ice Satellite Application Facility (OSISAF; ftp://osisaf.met.no/
archive/ice/conc, accessed on 18 November 2022). It is an operational product with the product ID of OSI-401-b (Tonboe
et al., 2017). The SSMIS is a polar orbiting conically scanning radiometer with the incidence angle around 50° and a swath
width of about 1700 km. It has window channels near 19, 37, 91, and 150 GHz and sounding channels near 22, 50, 60, and 183
GHz. The OSISAF SIC algorithm uses brightness temperature swath data as input, using the 19V, 37V and 37H channel data.
The brightness temperatures are corrected explicitly for air temperature, wind roughening over open water and water vapour
in the atmosphere prior to the SIC calculation. The algorithm uses dynamical tie-points based on the actual mean signatures
over the last 30 days. A hybrid algorithm is used which combines the Bootstrap (Comiso, 1986) and Bristol (Smith, 1996)
frequency mode algorithms. The results are analysed on the 10 km OSISAF grid. The SSMIS SIC is the main SIC product
assimilated in NEMO (Lellouche et al., 2022) and TOPAZ4 (Hackett et al., 2022). It is also assimilated in neXtSIM together
with the AMSR2 SIC (Williams et al., 2021).

### 240  3.1.3  NIS ice chart

Due to the large uncertainties in the passive microwave radiometers for low SIC conditions (e.g. Spreen et al., 2008), we
choose the NIS ice chart as an independent SIC product to evaluate the model simulations for sea ice edge and MIZ (section
5). It is from the CMEMS near real time product (https://doi.org/10.48670/moi-00128, accessed on June 2022). The ice chart





is produced based on manual interpretation of satellite data (Dinessen and Hackett, 2018), being a typical subjective analysis

product. Unlike the AMSR2/SSMIS SIC which only uses passive microwave measurements, the ice charting employs a variety of satellite observations to obtain a more realistic sea ice edge and MIZ. The main satellite data used are the weather independent Synthetic Aperture Radar (SAR) data from RadarSat-2 and Sentinel-1. The analyst also uses visual and infrared data from METOP, NOAA and MODIS in cloud free conditions. These satellites data cover the charting area several times a day and are resampled to 1 km grid spacing.

It is noted that the NIS ice chart is only provided during the working days, and only covers the European side of the Arctic. Nevertheless, the ice chart provides important details for the sea ice edge and MIZ. The NIS ice chart includes six ice types following the WMO Sea Ice Nomenclature (WMO, 2014): fast ice (SIC = 10/10), very closed drift ice ($9 - 10/10$), closed drift ice ($7 - 8/10$), open drift ice ($4 - 6/10$), very open drift ice ($1 - 3/10$), and open water ($0 - 1/10$). For practical use, a mean value is applied to denote the different ice classes in the ice chart (Dinessen and Hackett, 2018).

## 3.2 SIC analyses

The three SIC analyses are all from the CMEMS operational products, being the optimal estimates after data assimilation of the operational models NEMO, TOPAZ4 and neXtSIM. These analyses represent the state-of-the-art operational sea ice forecast products in Europe. New developments are being performed with the focus on improving model physics and spatial resolution, as well as extending for biogeochemical predictions.

### 3.2.1 NEMO SIC

The NEMO SIC analysis is from the CMEMS operational product (https://doi.org/10.48670/moi-00016, accessed May 2022). It is provided by Mercator Ocean of France through the Operational Mercator Global Ocean Analysis and Forecast System (Lellouche et al., 2022; Galloudec et al., 2022). The system uses version 3.6 of the NEMO model (Madec and the NEMO Team, 2017), with the sea ice component being the multi-category sea ice model LIM3 (Rousset et al., 2015). The system uses a

tripolar horizontal grid (Madec and Imbard, 1996), and a 50-level vertical grid with a decreasing resolution from 1 m at the surface to 450 m at the bottom. Its data assimilation system, SAM2 (Système d'Assimilation Mercator), uses a reduced-order Kalman filter derived from the singular evolutive extended Kalman (SEEK) filter (Brasseur and Verron, 2006). The assimilated observations include SIC, SIT, SST, in-situ T/S profiles, and sea level. The operational product includes daily and monthly mean files of temperature, salinity, currents, sea level, mixed layer depth and ice parameters over the global ocean. It also

includes hourly mean surface fields of sea level height, temperature and currents. The global ocean output files are displayed on a grid of 1/12 degree horizontal resolution with regular longitude/latitude equirectangular projection.

### 3.2.2 TOPAZ4 SIC

The TOPAZ4 SIC analysis is from the CMEMS operational product (https://doi.org/10.48670/moi-00001, accessed May 2022), which is the nominal product of the CMEMS Arctic Monitoring and Forecasting Center (MFC) for ocean physics (Hackett



et al., 2022). It is produced by the Arctic MFC through the operational TOPAZ4 Arctic Ocean and sea ice prediction system (Sakov et al., 2012), using the version 2.2.37 of HYCOM ocean model (Bleck, 2002) coupled to a one-category sea ice model with the EVP rheology (Hunke and Dukowicz, 1997). Its sea ice thermodynamics is described in Drange and Simonsen (1996), with a correction of heat flux for sub-grid scale ice thickness heterogeneity following Fichefet and Morales Maqueda (1997). The model domain covers the North Atlantic and Arctic basins, and grid spacing approximately 12-16 km. The model uses 28 hybrid vertical layers, with the vertical coordinate as isopycnal in the stratified open ocean and z-coordinate in the unstratified mixed layer. A 100-member deterministic EnKF is used in TOPAZ4 for assimilation of SIC, SIT, sea ice drift, SST, SSS, sea level anomaly, and in-situ T/S profiles (Hackett et al., 2022). The analysis is then interpolated and disseminated to a 12.5 km × 12.5 km grid using the polar stereographic projection.

### 3.2.3 neXtSIM SIC

The neXtSIM SIC is from the CMEMS operational product (https://doi.org/10.48670/moi-00004, accessed May 2022). It is a hourly product produced by the Arctic MFC through the neXtSIM sea ice prediction system (Hackett et al., 2022). The neXtSIM is a stand-alone sea ice model using the Brittle-Bingham-Maxwell sea ice rheology (Rampal et al., 2019). The model domain covers the central Arctic, excluding the Canadian Archipelago and the Pacific side of Bering Strait (Williams et al., 2021). The model uses a Lagrangian triangular mesh with the mesh element size approximately 7.5 km from point to the opposite side variable (the side length is approximately 10 km). A remeshing procedure is used to avoid anomalously deformed meshes. The model is forced with surface atmosphere forcing from the ECMWF and ocean forcing from TOPAZ4. The model uses the direct insertion method for assimilation, with the observed SIC from a weighted SSMIS SIC and AMSR2 SIC and a fixed model SIC uncertainty of 30% (Williams et al., 2021). The assimilation is run daily before the forecast is launched. The output variables are hourly SIC, SIT, sea ice drift velocity and snow depth. The adaptive Lagrangian mesh is interpolated for convenience to a 3 km resolution regular grid using the polar stereographic projection.

### 3.3 Weekly mean CS2SMOS SIT

The weekly mean CS2SMOS SIT is from ESA (ftp://smos-diss.eo.esa.int/SMOS/L4_SIT/L4/north/, accessed 7 October 2022). It is a weighted mean of the weekly mean SMOS thin SIT (Tian-Kunze e et al., 2014) and the weekly mean CryoSat-2 SIT (Laxon et al., 2013), with the spatially no observation area interpolated from the surrounding observations using the OI method (Ricker et al., 2017). This weekly averaged product is generated every day at the Alfred Wegener Institut (AWI). The data is available from November 2010, but only available in the winter seasons (from mid-October to mid-April). The data are projected onto the 25 km EASE2 Grid, based on a polar aspect spherical Lambert azimuthal equal-area projection (Ricker et al., 2017). This CS2SMOS SIT is used in section 4 for evaluating the effect of SIC assimilation on the SIT simulation.





### 3.4 OSTIA skin SST

The Operational Sea Surface Temperature and Ice Analysis (OSTIA) diurnal skin SST product is from CMEMS (https://doi.org/10.48670/moi-00167, accessed on 5 September 2022). It is an hourly mean skin SST at $0.25° \times 0.25°$ horizontal resolution, analyzed by the UK Met Office, using in-situ and satellite data from infra-red radiometers (While and Martin, 2013). The skin SST is the temperature measured by satellite infra-red radiometers and can experience a large diurnal cycle. The skin SST L4 product is created by combining: 1) the OSTIA foundation SST analysis which uses in-situ and satellite observations; 2) the

OSTIA diurnal warm layer analysis which uses satellite observations; and 3) a cool skin model. This product is used in section 4 for assessing the effect of SIC assimilation on the SST simulation.

### 3.5 ISAS SSS

The In Situ Analysis System (ISAS) SSS is also from CMEMS (https://doi.org/10.48670/moi-00037, access on 15 November 2022). It is a quality-controlled gridded salinity field based on the objective analysis (OA) of the near real time in-situ salinity

measurements from the Coriolis Database (Szekely and Dobler, 2022). The measurements used a variety of instruments, mainly Argos, moorings, drifting buoys and sea mammals. The OA is performed monthly using the ISAS tool (Gaillard et al., 2016). The contribution of each quality-controlled salinity observation will firstly be assessed relative to a first guess (climatology). The ISAS SSS is then reconstructed by summing the objectively analyzed anomalies and the first guess field. It it noted that the analyzed results could be very close to the climatology in the poorly sampled areas (Szekely and Dobler, 2022). Nevertheless,

it remains one of the best SSS data in the Arctic, particularly under the sea ice. This product is used in section 4 for evaluating the effect of SIC assimilation on the SSS simulation.

## 4 Effect of LAON assimilation

The coupled HYCOM-CICE model is initialized and spun up from 2010 using the World Ocean Atlas (WOA) 2018 climatology. A twin experiment with and without the LAON data assimilation is performed to evaluate the effect of the LAON

assimilation on the model simulations, from 1 January 2021 to 30 April 2022. In the following, the experiments with and without data assimilation are denoted as LAON and Free_run, respectively.

Anomaly Correlation Coefficient (ACC) and Root Mean Squre Error (RMSE) are two common metrics used for model evaluations. However, ACC tends to produce ambiguous results. As a simple example, let us assume that we have three anomaly series against the climate: an observed anomaly series $O = [1, 2, 3, 2, 1]$, model A predicted anomaly series $A = [1, 2, 3, 4, 5]$, and

model B predicted anomaly series $B = [10, 20, 30, 20, 10]$. Even from simple judgement we can know that model A performs better than model B. However, the ACC between O and A is 0, while it is 1 between O and B. This means a higher ACC does not represent a better performance, indicating a clear disrepancy in the ACC method. As a result, we here will use the RMSE for evaluation. Denote $\mathbf{O}(O_k)$ as the observed variable vector (e.g. SIC, SST, SSS etc), $\mathbf{M}(M_k)$ as the corresponding model





variable vector, we have

$$RMSE = \sqrt{\frac{1}{m}\sum_{k=1}^{m}(M_k - O_k)^2} \tag{21}$$

where $m$ is the total model grid number (here the observations have been interpolated to the model grid). In addition, we use bias as a supplementary metric, which is a simple mean difference between the model and observations.

| (a) Free_run Jun 2021 | (b) Free_run Sep 2021 | (c) Free_run Dec 2021 | (d) Free_run Mar 2022 |
| (e) AMSR2 Jun 2021 | (f) AMSR2 Sep 2021 | (g) AMSR2 Dec 2021 | (h) AMSR2 Mar 2022 |
| (i) LAON Jun 2021 | (j) LAON Sep 2021 | (k) LAON Dec 2021 | (l) LAON Mar 2022 |

**Figure 4.** Monthly mean SIC from Free_run simulations (upper), AMSR2 observations (middle), and LAON simulations (lower).





## 4.1 SIC, SIE and SIA

Figure 4 compares the monthly mean SIC between the Free_run (upper), AMSR2 observations (middle) and the LAON simu-
lation (lower), with the columns from left to right showing the dates June 2021, September 2021, December 2021 and March
2022, respectively. To remove the effect of data assimilation during the early stage, June 2021 is chosen as the starting month
for analysis. On the whole, the Free_run simulates the SIC quite well during the winter months (c, d vs. g, h in Fig. 4), except
for some areas near the sea ice edge. However, there are considerable biases in the simulated SIC during the summer months
(a, b vs. e, f in Fig. 4). In particular, the simulated September sea ice cover deviates considerably from the observation (b vs. f
in Fig. 4). By contrast, the LAON assimilation significantly improves the simulation (i−l vs. e−h in Fig. 4). The spatial pattern
of sea ice cover is particularly well reproduced, for example, the low SIC areas in the Beaufort Sea in September 2021 (j vs. f
in Fig. 4), in the Hudson Bay in December 2021 (k vs. g in Fig. 4), and in the Nansen Basin (north of Spitsbergen) in March
2022 (l vs. h in Fig. 4).

    The daily mean bias and RMSE of the simulated SIC are shown in Fig. 5a. As partly demonstrated in Fig. 4, the Free_run
SIC has large RMSE (about 0.3) and relatively low bias during the summer season, indicating large spatial mismatch during
the summer season. The low daily bias is mainly due to the offset between the overestimate and underestimate of the SIC in
different areas in the Arctic (see also Fig. 4b vs. f). By contrast, the LAON SIC has a much lower daily mean bias and RMSE,
with the mean values being 0.006 and 0.066 for the whole period.

    SIE and SIA are two derivatives of SIC. In this section, SIE is defined as the total area where SIC is $\geq 0.15$, whereas SIA
is defined as the total area covered by ice. The simulated (Free_run and LAON) daily SIE and SIA are compared with the
AMSR2 observations in Fig. 5b, with their mean bias shown in Fig. 5c. It is seen that all the simulated winter SIE and SIA
agree very well with the AMSR2 observations. However, there are large biases in the Free_run SIE and SIA during the summer
season. The mean bias of SIE reaches over $1.0 \times 10^6$ km$^2$ during July to October 2021, being about 20% of the observed SIE
(see Fig. 5b). The Free_run SIA has a large variation from notably less to significantly larger than the observations as indicated
by the daily mean bias and RMSE (Fig. 5c). On the contrary, the mean biases of the LAON SIE and SIA are generally close to
$0.2 \times 10^6$ km$^2$, which is about 5% of the total SIE and SIA.

## 4.2 SIT and SIV

SIT is not assimilated in the model. This implies that the model SIT will likely deviate from the observation if it was initially
biased. To be consistent with the observed weekly mean CS2SMOS SIT, the modeled SIV and SIT have also been averaged
weekly. As the CICE model only tracks SIV, the model SIT is calculated here via SIV/(SIC+$10^{-16}$) for each model grid. The
observed SIV is calculated via CS2SMOS SIT × AMSR2 SIC for each grid.

    Figure 6 compares the SIT between the Free_run and LAON simulations to the CS2SMOS observations, with the columns
from left to right showing the dates $1-7$ January 2021, $9-15$ April 2021, $15-21$ October 2021 and $1-7$ January 2022,
respectively. It is seen that the model initial SIT fields are considerably biased, with a much larger area of thick multiyear ice
(MYI) located in the Beaufort Sea and north of the Canadian Archipelago (Fig. 6a and i), whereas the observed MYI is mainly

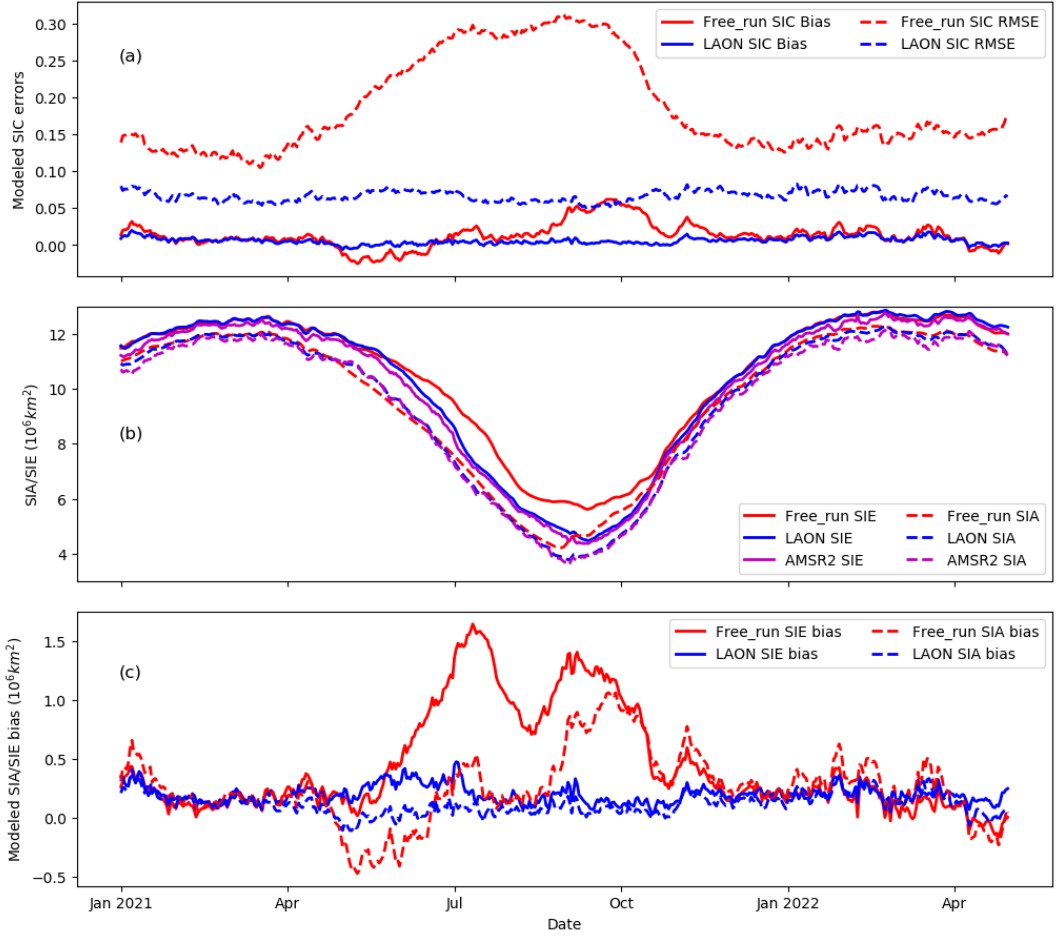

**Figure 5.** Statistics of the simulated SIC, SIE and SIA: a) Modeled daily SIC mean bias and RMSE, b) Modeled and observed SIE/SIA, and c) Modeled SIE/SIA mean bias.

located north of the Canadian Archipelago and Greenland (Fig. 6e). There is only mild difference in the Free_run and LAON simulations during the winter period until 9 − 15 April 2021 (see Figs. 6b and j), both showing the MYI in the Beaufort and East Siberia Seas, while the the observations show the MYI is mainly located north of the Canadian Archipelago and Greenland (Fig. 6f).

Since the CS2SMOS SIT is only available during the winter period from mid-October to mid-April, we here choose the mid-October 2021 to examine the SIT after the summer season (Figs. 6c, g and k). The overestimated September SIC in the East Siberian Sea in the Free_run simulation (Fig. 4b vs. f) results in significant MYI there in October (Fig. 6c). By contrast, in the LAON simulation most of the MYI in the East Siberian Sea has been replaced with first-year ice (FYI). The MYI are now located mostly to the north of the Canadian Archipelago and Greenland (Fig. 6k), although the thickness is considerably

overestimated compared with the CS2SMOS observation (Fig. 6g). It is particularly noteworthy that the LAON simulation





produces a much more reasonable spatial SIT pattern in the early 2022 (Fig. 6l vs. h), whereas the Free_run simulation returns to a similar pattern as one year before (Fig. 6d vs. a).

The initial SIV is about 28.2% overestimated (Fig. 7a), which is corresponding to about 0.23 m thicker than the observation (Fig. 7b). The Free_run SIV is unanimously overestimated when compared with the observation during the winter period 385 (CS2SMOS SIT not available during the summer period), with a general increasing bias with time from November 2021 until April 2022. The SIC assimilation successfully reduces the SIV bias, which is already close to the observation (Fig. 7a). Such improvement can also be clearly seen in the SIT mean bias, where the overall mean bias has dropped to about 0.12 m in

(a) Free_run 1-7 Jan 2021  (b) Free_run 9-15 Apr 2021  (c) Free_run 15-21 Oct 2021  (d) Free_run 1-7 Jan 2022

(e) CS2SMOS 1-7 Jan 2021  (f) CS2SMOS 9-15 Apr 2021  (g) CS2SMOS 15-21 Oct 2021  (h) CS2SMOS 1-7 Jan 2022

(i) LAON 1-7 Jan 2021  (j) LAON 9-15 Apr 2021  (k) LAON 15-21 Oct 2021  (l) LAON 1-7 Jan 2022

**Figure 6.** Weekly mean SIT from Free_run simulations (upper), CS2SMOS observations (middle), and LAON simulations (lower).





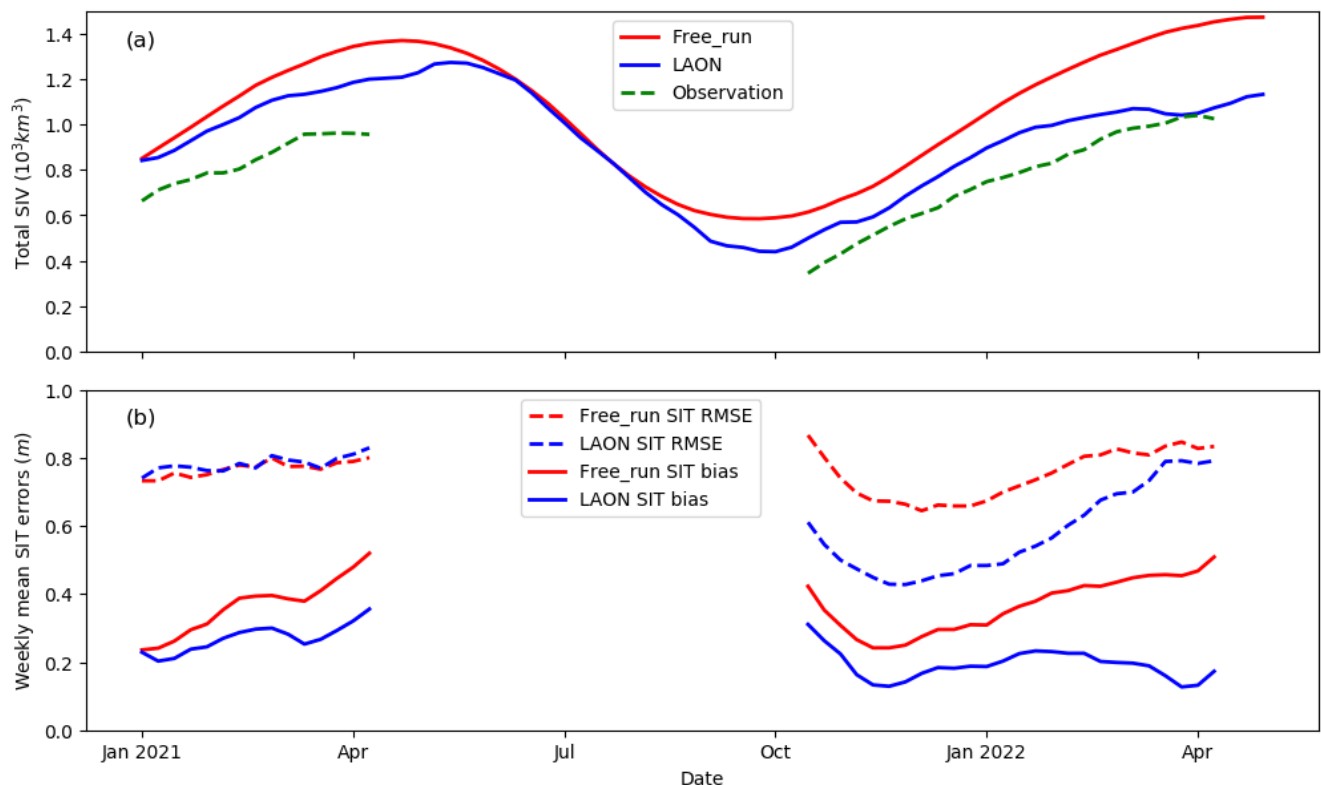

**Figure 7.** Statistics of the simulated SIT and SIV: a) Total SIV, and b) Modeled SIT mean bias and RMSE.

April 2022 (Fig. 7b). It is anticipated that, after several years' SIC assimilation, the overall SIT spatial distribution would have more significant improvement during the summer season. Nevertheless, a direct SIT assimilation would be more effective and
prompt.

The SIT RMSE remains similar between the Free_run and LAON simulations during the early stage from January to April 2021 (Fig. 7b), and differs significantly after the summer period (Fig. 7b). However, the LAON SIT RMSE tends to increase more rapidly than the Free_run after November 2021 (Fig. 7b). A detailed check of the sea ice velocity field shows that the exceptional thick ice north of the Canadian Archipelago and Greenland (see Fig. 6l) considerably affected the sea ice
circulation, which is accumulated with time and thus results in large bias in the SIT fields in the LAON simulations. This problem will be further investigated in a following study with additional assimilation of SIT.

### 4.3 SST and SSS

SST is a challenging parameter to define precisely, since the upper ocean has a complex and variable vertical temperature structure related to ocean turbulence and air-sea fluxes of heat, moisture and momentum. For comparison, we use the top layer
model SST, which is the mean temperature of the top 1 m. The hourly OSTIA skin SST is also averaged to obtain the monthly

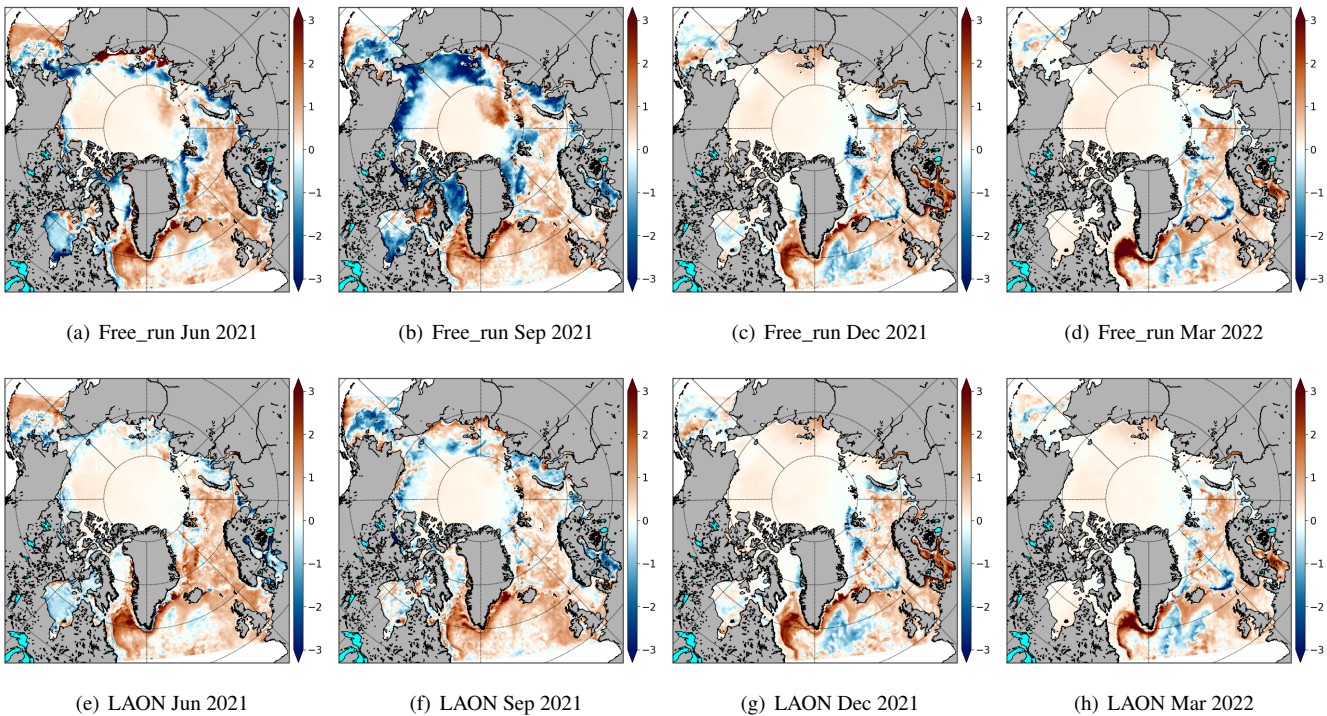

(a) Free_run Jun 2021     (b) Free_run Sep 2021     (c) Free_run Dec 2021     (d) Free_run Mar 2022

(e) LAON Jun 2021     (f) LAON Sep 2021     (g) LAON Dec 2021     (h) LAON Mar 2022

**Figure 8.** Monthly mean SST bias from Free_run simulations (upper), and LAON simulations (lower). The results are evaluated against the OSTIA monthly mean skin SST from CMEMS.

mean SST. Figure 8 compares the monthly mean SST bias between the Free_run and LAON simulations evaluated against the OSTIA monthly mean skin SST. It is noteworthy that the model SST has a positive bias in much of the ice-free area, with the most significant bias around the northern Labrador Sea and southern Davis Strait. Consistent with the SIC (Fig. 4), the SIC assimilation only slightly improved the SST simulations during the winter season (c, d vs. g, h in Fig. 8), mainly close to the

ice edge. However, during the summer season, the SIC assimilation considerably improved the simulated SST. The Free_run simulation produced a large negative bias in the East Siberian, Chukchi, Beaufort and Greenland Seas, and a large positive bias north of the Laptev Sea (Fig. 8b). These biases are considerably mitigated in the LAON simulation (8f). It is also noteworthy that in the Baffin and Hudson Bays the marked biases in the Free_run are also considerably mitigated in the LAON simulation (a, b vs. e, f in Fig. 8).

The SIC assimilation generally has minor effect on the SSS (Fig. 9). Over the large ice-free area in the North Atlantic Ocean, both simulations with and without the assimilation reproduce the SSS very well, with the SSS bias generally smaller than 0.5 PSU. However, there is significant SSS bias in the Arctic. In particular, the SSS is considerably overestimated in much of the Arctic shelf seas, with occasional large bias even up to 30 PSU. Such large overestimates are likely related to the external forcing of river discharges and inaccurate evaporation-precipitation processes. In addition, there is clear SSS bias in the central

Arctic under the sea ice. This tends to suggest that the SSS in the Arctic proper has accumulated a substantial drift during the

**Figure 9.** Monthly mean SSS bias from Free_run simulations (upper), LAON simulations (middle), and ISAS SSS STD (lower), where STD denotes standard deviation. The results are evaluated against the monthly mean ISAS SSS from CMEMS.

10-year spin-up run, as no nudging to the climate is performed under the sea ice. It is noteworthy that these large bias areas are generally collocated with the larger uncertainty areas in the observed ISAS SSS (i−1 in Fig. 9). Further work is needed to clarify the uncertainties and mitigate the deficiencies.

The melt water from the Greenland ice sheet is not included in the current river discharge. This would induce overestimate
of SSS in the Baffin Bay and Davis Strait, but counteracted by the extra sea ice in the Free_run. The SIC assimilation tends to remove the extra sea ice in September (Fig. 4j), thus recovering the slight overestimate of SSS in the Baffin Bay and Davis Strait (Fig. 9f).





# 5    Comparison with other products

SIC is so far one of the most extensively observed sea ice parameters from space. The microwave radiometers such as the

AMSR2 and SSMIS have the capability of penetrating clouds and continuously monitoring the sea ice throughout the year. As a result, they are widely used for sea ice monitoring and data assimilation. However, passive microwave radiometers tend to underestimate low SIC (Spreen et al., 2008; Ozsoy-Cicek et al., 2009). By contrast, the manually analyzed ice chart based on a much larger data set is more reliable for accurately detecting low SIC areas and sea ice edge (Ozsoy-Cicek et al., 2009; Breivik et al., 2009; Posey et al., 2015). In this section, we evaluate the LAON assimilation by comparing the NorHAPS simulations to

the CMEMS operational SIC analyses and the passive microwave radiometer observations, evaluated against the NIS ice chart.

## 5.1    Daily SIC spatial distribution

Figure 10 compares the daily SIC of the NorHAPS LAON with six other daily products for 16 March 2022. Three are from the CMEMS model analyses, namely NEMO, TOPAZ4 and neXtSIM, and the other three are from observations, namely AMSR2, SSMIS and NIS ice chart. All the SICs have been interpolated to the NIS ice chart grid. The mid-March is chosen as a typical

winter condition. For better demonstration of the ice edge, the areas where SIC < 0.1 have been removed. It is noteworthy that NorHAPS LAON assimilated the AMSR2 SIC, NEMO and TOPAZ4 assimilated the SSMIS SIC (Lellouche et al., 2022; Hackett et al., 2022), and neXtSIM assimilated a combined AMSR2 and SSMIS SIC (Williams et al., 2021). The NIS ice chart is not assimilated by any of the models and is therefore an independent observation.

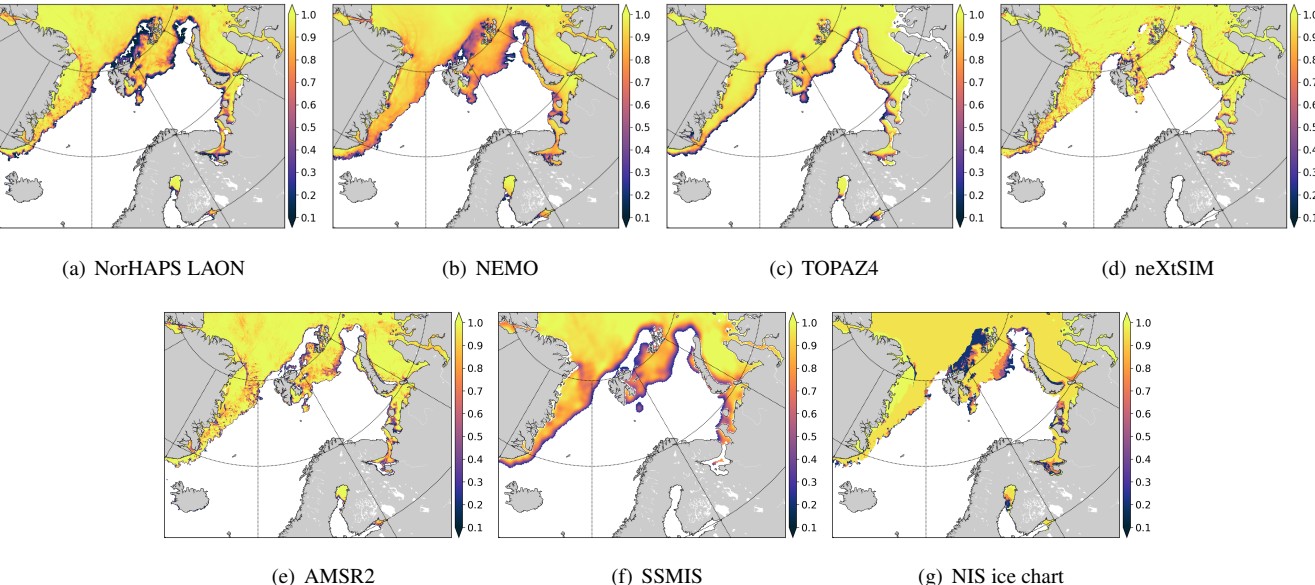

**Figure 10.** Daily SIC on 16 March 2022 from different model analyses (upper), and observations (lower). For better illustration of the ice edge, the areas where SIC < 0.1 have been removed.





As mentioned above, the passive microwave radiometers generally have quite large uncertainties in the low SIC area. This
can also be seen in both AMSR2 (Fig. 10e) and SSMIS (Fig. 10f), where the very open drift ice north of Svalbard seen in the
NIS ice chart (Fig. 10g) is missing. The four models generally reproduced most of the sea ice features in the European Arctic
(Fig. 10a-d). However, there are large differences among the four models in the simulation of the very open drift ice area in the
Nansen Basin (north of Svalbard and Franz Josef Land). When compared with the NIS ice chart (Fig. 10g), the NEMO SIC
analysis (Fig. 10b) is moderately overestimated, showing an area of open drift ice and close drift ice. TOPAZ4 and neXtSIM
tend to markedly overestimate the SIC in this very open drift ice area, both showing a large part of very close drift ice (see
Fig. 10c, d). By contrast, NorHAPS LAON (Fig. 10a) produced a close simulation to the observed NIS ice chart. The overall
seasonal evolution is assessed through the integrated ice edge error (IIEE) and the integrated MIZ error (IME) below.

## 5.2    IIEE and IME

The IIEE is determined following Goessling et al. (2016)

$$IIEE = \int_A max(c_f - c_t, 0)dA + \int_A max(c_t - c_f, 0)dA \quad (22)$$

where $A$ denotes the whole model domain, the subscripts $f$ and $t$ denote the estimate and the truth (here we use the NIS ice
chart as an approximate). The first term on the right side of Eq. (22) denotes the overestimate, and the second term denotes
the underestimate. In Goessling et al. (2016), the variable $c = 1$ where SIC $> 0.15$ and $c = 0$ elsewhere. The demarcation
value 0.15 is commonly used in the sea ice and climate modeling communities as the sea ice edge. However, this is rather
arbitrary as there is no special reason to use 0.15 rather than e.g. 0.10 as the sea ice edge. In fact, WMO has used 0.10 as the
demarcation between Open Water and Very Open Ice since 1970 (WMO, 2014), which has long been applied in the sea ice
charting community, such as the NIS and US National Ice Center ice charts. Therefore, using 0.10 as the demarcation for sea
ice edge would be more consistent and helpful for the joint sea ice modeling and observation community. In this section, we
use 0.10 as the demarcation for the ice edge.

Following the formulation for the IIEE, we define the IME as

$$IME = \int_A max(c_f - c_t, 0)dA + \int_A max(c_t - c_f, 0)dA \quad (23)$$

The only difference between Eq. (23) and Eq. (22) is the definition of the variable $c$. For the IME, $c = 1$ where SIC $\in [0.1, 0.8]$
and $c = 0$ elsewhere.

Figure 11 compares the IIEE and IME of NorHAPS LAON (shown as NorHAPS) with two satellite observations (AMSR2
and SSMIS), and three model analyses (NEMO, TOPAZ4 and neXtSIM). All the data are evaluated against the NIS ice chart
(total valid data number of 335). The discontinuities in the IIEE and IME are due to the fact that the NIS ice chart is only
available on working days. The Baltic Sea is removed in all the calculations, as neXtSIM does not cover this area. For other
areas, if a certain product has no data while the NIS ice chart has a SIC $> 0.8$, then no IIEE and IME is accounted. This
treatment is to remove the coastal effect on other products. On the whole, the IME is about twice of the IIEE, indicating that



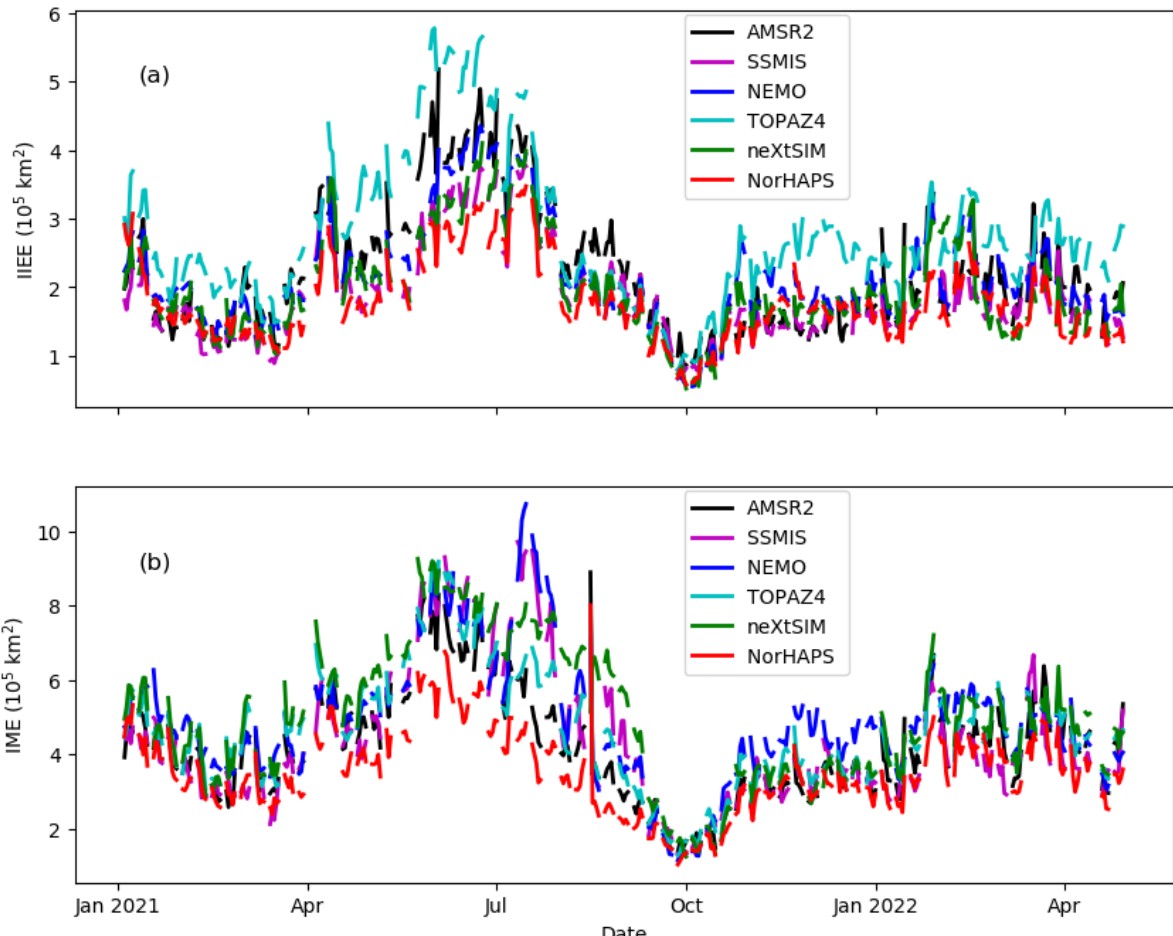

**Figure 11.** Integrated ice edge error (IIEE) and integrated MIZ error (IME) of different SIC products evaluated against the Norwegian ice chart from 1 January 2021 to 30 April 2022: (a) IIEE, and (b) IME.

modeling MIZ is considerably more difficult than modeling sea ice edge. It is surprising to see that late September to early October is a time of the lowest IIEE and IME, whereas late June to early July is the time of largest IIEE and IME, which are ubiquitously in all the observations and model analyses.

Table 1 summarizes the statistics of the IIEE and IME over the whole period. It is seen that the two observation products AMSR2 and SSMIS SIC have different capabilities in describing the sea ice edge and MIZ. SSMIS has higher capability in

capturing the sea ice edge, whereas AMSR2 has higher capability in describing the MIZ. The CMEMS analyses (NEMO, TOPAZ4 and neXtSIM) generally have larger IIEE and IME than the SSMIS. TOPAZ4 has a markedly larger bias in the simulated ice edge (also see Fig. 11a), however it has a better simulation of the MIZ than NEMO and neXtSIM (Fig. 11b). On



**Table 1.** Statistics of IIEE and IME for the different SIC products evaluated against the NIS ice chart. The units of the mean and std of the IIEE and IME are $10^5$ km$^2$, in which std denotes standard deviation. The p value denotes the probability assuming that the concerned product does not have a statistically different IIEE/IME from NorHAPS.

| Product | IIEE | | | IME | | |
|---|---|---|---|---|---|---|
| | mean | std | p value | mean | std | p value |
| AMSR2 | 2.22 | 0.91 | $8.04 \times 10^{-13}$ | 4.20 | 1.44 | $5.47 \times 10^{-11}$ |
| SSMIS | 1.89 | 0.68 | $3.02 \times 10^{-2}$ | 4.39 | 1.82 | $1.07 \times 10^{-12}$ |
| NEMO | 2.16 | 0.77 | $2.00 \times 10^{-12}$ | 4.89 | 1.70 | $1.51 \times 10^{-30}$ |
| TOPAZ4 | 2.74 | 1.03 | $5.54 \times 10^{-41}$ | 4.62 | 1.56 | $5.44 \times 10^{-23}$ |
| neXtSIM | 1.99 | 0.73 | $6.91 \times 10^{-5}$ | 4.99 | 1.79 | $3.12 \times 10^{-32}$ |
| NorHAPS | 1.78 | 0.59 | 1.0 | 3.54 | 1.11 | 1.0 |

the contrary, neXtSIM has a relatively small bias in the simulated ice edge, but has a large bias in the simulated MIZ, indicating an overestimate of the simulated SIC in the MIZ. This is consistent with the spatial distribution in Fig. 10.

The NorHAPS LAON (shown as NorHAPS) has a significantly lower IIEE and IME than all the other products (Fig. 11b, Table 1). Using Welch's unequal variances t-test (Welch, 1947), we have estimated the p value (Table 1) which denotes the probability assuming a concerned product (AMSR2, SSMIS, NEMO, TOPAZ4 or neXtSIM) does not have a statistically different mean IIEE/IME from NorHAPS. It is seen that all the p values are far smaller than 0.01, except SSMIS having a p value of about 0.03 for the IIEE. It is not surprising that the p values for the NorHAPS IIEE and IME are both 1.0, since they
compare with the own data themselves. It is particularly noteworthy that the NorHAPS LAON produces a significant lower IIEE and IME than the observation AMSR2 SIC that is assimilated in NorHAPS, with the p values both lower than $1.0 \times 10^{-10}$ for the IIEE and IME (Table 1). On the whole, the improvement is especially pronounced during the summer season (Fig. 11).

### 5.3    Hourly evolution

Operational sea ice forecast rarely provides hourly product, and neXtSIM in CMEMS is one exception. For comparison, we
selected two points close to the sea ice edge. One is in the Fram Strait (0°E, 80°N) in April 2021, and the other in Nansen Basin (30°E, 81°N) in March 2022, both covering one month (Fig. 12). The daily NIS ice chart, AMSR2 SIC and SSMIS SIC are also added for reference. All the data are interpolated to the two points using the nearest-neighbour method. It is seen that the daily AMSR2 SIC generally has a better agreement with the NIS ice chart than the SSMIS SIC. This is partly due to the relatively coarse spatial resolution of the SSMIS SIC, which could have a smoothing effect near the ice edge, averaging high
and low SIC.

There are significant differences in the hourly SIC between neXtSIM, Free_run and LAON. On the whole, neXtSIM tends to overestimate the SIC in the MIZ, as already mentioned before (Figs. 10 and 11). Both neXtSIM and Free_run significantly overestimate the SIC during 10−16 March 2022, simulating the very open ice as very close ice. By contrast, the NorHAPS
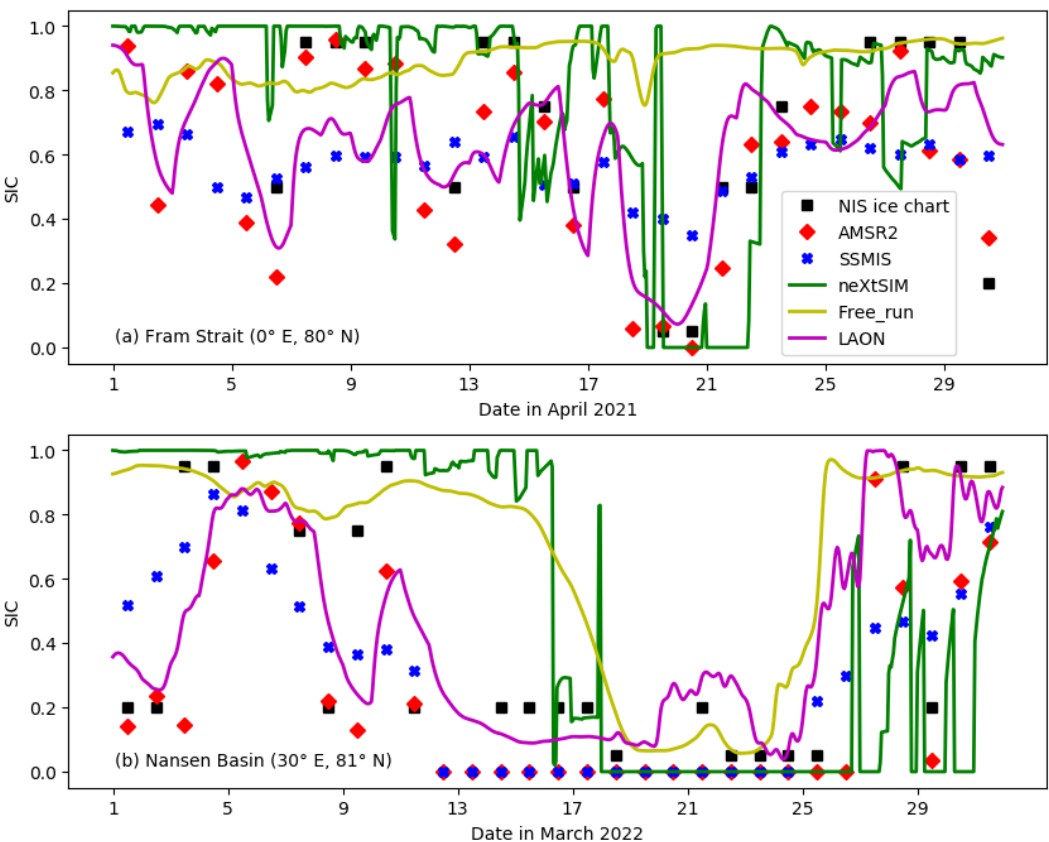

**Figure 12.** Comparison of neXtSIM and NorHAPS hourly SIC close to the sea ice edge: (a) Fram Strait point (0° E, 80° N) in April 2021; (b) Nansen Basin point (30° E, 81° N) in March 2022. The NIS ice chart, AMSR2 and SSMIS SIC are daily and added for reference. All the data are interpolated to the corresponding locations using the nearest neighbour method.

LAON successfully reproduces the evolution of the very open ice as classified by the NIS ice charts. In addition, the NorHAPS

LAON also produces a continuous and smooth change in the local SIC variation. On the contrary, neXtSIM tends to produces abrupt changes in the local SIC (often jumping between 0 and 1 immediately). Such abrupt changes could partly be explained by the physical processes (e.g. damage-induced rapid deformation) in neXtSIM. However, opening or closing of sea ice leads over 3 km wide in 1 hour is generally unlikely (considering the neXtSIM spatial resolution of 7 km and interpolated to 3 km grid). More sub-daily observations of the MIZ are needed to clarify such variations.



## 6 Discussion

### 6.1 Model and observation uncertainties

The overall goal of data assimilation is to find the optimal estimate of the concerned variables, as closer as possible to the true values. Such optimal estimate is generally a weighted average of the model simulations and the observations, with the weights commonly being proportional to the inverse of the error covariance of the model and the observations. Therefore, the uncertainties of the model and the observations provide essential information for data assimilation and need to be seriously treated.

In sea ice observations, uncertainty has gradually become a standard part in the operational products, e.g. SMOS SIT (Tian-Kunze et al., 2014), weekly mean CS2SMOS SIT (Ricker et al., 2017), and OSISAF SSMIS SIC (Tonboe et al., 2017; Lavergne et al., 2019). Such estimated uncertainties provide very useful information of the observed sea ice parameters, thus highly valuable for data assimilation.

Estimate of model uncertainty remains one of the most difficult part in data assimilation. The EnKF provides a feasible way to estimate this uncertainty. However, due to the non-Gaussian distribution of SIC, applying EnKF for SIC uncertainty estimate is very challenging. It can easily be biased, or even collapse into zero uncertainty in the winter central Arctic (e.g. Lisæter et al., 2003; Fritzner et al., 2018). In the current LAON assimilation, we have used a very simple formulation (Eq. 13) to approximate the model uncertainty. The results indicate that this simple equation may have captured the essential part of the model uncertainty.

The local covariance assumption in the current LAON assimilation may lose some useful information during the assimilation. However, the heterogeneous spatial distribution of the Arctic sea ice cover, particularly the existence of the sea ice edge, may favor such a very local covariance (simplified as variance). In EnKF assimilation such as the operational TOPAZ4 system, the covariance is calculated from the previous one-week, which at the ice edge can cause noticeable mismatch when applying to the new analysis. The high resolution, local covariance, concurrent model and observation uncertainties, and continuous assimilation in the LAON method are likely the main reasons for a better simulation of the ice edge and MIZ.

### 6.2 Data accuracy vs. independence

With the continuous development in the remote sensing technique, it becomes more and more common to have multiple observations for the same sea ice variables or parameters. Such multiple data may have different coverage, resolution and accuracy. An important issue arising from this situation is how to best use the observations in data assimilation and model evaluation, or whether we have some criteria to determine the data utilization during data assimilation and model evaluation.

There has been little disputation in selecting an appropriate data for data assimilation or model evaluation alone. In general, it is preferable to select the observation data with larger spatial coverage and higher accuracy. For the spatial resolution, a common practice is to choose the observations having the closest resolution to the model, although other resolution data are also utilized. There also seems an implicit agreement in the research community that the data accuracy for model evaluation





should not be lower than that for data assimilation, whereas higher resolution and larger coverage are generally preferable but not indispensable. In addition, data independence is often stressed for model evaluation.

According to the timeliness, model evaluation can be separated into two types: analysis evaluation and prediction evaluation.
Prediction evaluation can basically be seen independent of data assimilation. Consider a general data assimilation and prediction evaluation loop: 1) data assimilation of earlier observations, 2) model prediction, and 3) prediction evaluation with new observations. Since the new observations are independent of the earlier observations, there is no inherent conflicts between data assimilation and prediction evaluation in terms of independence.

Unlike the prediction evaluation, there remains large disagreement on data usage for data assimilation and analysis eval-
uation in terms of accuracy and independence. When independent more accurate data are available (generally through other instruments with limited temporal and spatial coverage), these data can be readily applied for analysis evaluation. However, when such data is not available, we have to consider other observations, including those that have already been assimilated. As a common practice, data assimilation is performed earlier than analysis evaluation, where the more accurate data will generally be firstly used for data assimilation. In this case, no consensus has been reached whether to preferably use the same more
accurate data or to use another independent but less accurate data for analysis evaluation. In data science, evaluating model performance with the data used for training is generally not acceptable, because it can easily generate overoptimistic and overfitted models. This is reasonable as the trained model is heavily based on the data. However, data assimilation has essential difference from the model training. In particular, data assimilation does not change the model itself (not for model parameter estimation). Using the same data for analysis evaluation does not necessarily lead to overoptimistic, as using other data would
have other even more severe limitations. Since the ultimate goal of data assimilation is to provide the optimal estimate, even the application of cross-validation method for analysis evaluation is not encouraged, as it would lose some optimality by reserving the data from data assimilation. One extreme case is perhaps quite intuitive: with a true value (100% accuracy), it would be best both for data assimilation and model evaluation. A low-accuracy independent data remains difficult to provide a convincing evaluation. Therefore, accuracy should have a higher priority than independence for analysis evaluation. This again stresses
the importance of estimation of observation uncertainty, which can be seen as a qualitative description of accuracy.

In the present study, we have used two SIC data for model evaluation: AMSR2 SIC and NIS ice chart. The AMSR2 SIC is assimilated in the present study and is therefore closely related to the model SIC analysis. It is used for SIC evaluation in section 4. The NIS ice chart is an independent observation for evaluating the model SIC analysis, and it is used in section 5 for evaluating the sea ice edge and MIZ. The main reason for such a distinction lies in that the NIS ice chart provides a
more accurate observation of sea ice edge and MIZ, which are often underestimated in the passive microwave radiometer observations (e.g. Spreen et al., 2008; Ozsoy-Cicek et al., 2009; Breivik et al., 2009; Posey et al., 2015). On the contrary, the coarse resolution of the NIS ice chart in the SIC space tends to provide a very rough estimate of SIC, so the AMSR2 SIC would be more accurate for SIC evaluation. This suggests that observations may have different accuracy for different model variables, which is helpful to be considered during data assimilation and model evaluations.





# 7 Conclusions


In this paper, we have introduced the theory of LAON for data assimilation, which is designed to gradually nudge the model value to the optimal estimate. It is applied here for assimilating the AMSR2 SIC into the multi-category CICE model in the high-resolution pan-Arctic coupled ocean and sea ice modeling and prediction system NorHAPS. A twin experiment with and without the LAON assimilation is performed, and the results are thoroughly evaluated against a variety of sea ice and ocean

observations as well as three CMEMS SIC analyses. Based on the model evaluation, we have the following conclusions:

— The LAON assimilation of SIC greatly improves the simulation of SIC and its derivatives SIE and SIA. The LAON SIC has a low mean RMSE of about 0.066 for the whole period, whereas the Free_run SIC has a much higher RMSE, being about 0.15 during the winter season and 0.3 during the summer season. The LAON assimilation significantly improves the simulated sea ice edge and MIZ evaluated against the NIS ice chart. It produces a significantly lower IIEE and IME than the two passive

microwave radiometer observations AMSR2 and SSMIS, as well as the three CMEMS SIC analyses NEMO, TOPAZ4 and neXtSIM which uses EnKFs and direct insertion method for data assimilation. The LAON also produces a continuous evolution of the simulated SIC, which provides a realistic description for sub-daily SIC evolution with daily observations.

— The LAON assimilation of SIC improves the simulation of SIT and SIV. The spatial pattern of the simulated SIT is noticeably improved after a one-year LAON assimilation. The LAON assimilation also reduces the overestimate of SIV, with

the bias in the second year being less than 150 km$^3$ compared with over 200 km$^3$ in the Free_run simulation (Fig. 7). However, the large SIT RMSE suggests that the LAON assimilation of SIC is unlikely to fully correct the SIT bias and therefore a direct assimilation of the SIT is highly needed.

— The LAON assimilation of SIC improves the SST simulation. In particular, the LAON assimilation considerably mitigates the summer large SST bias in the East Siberian, Chukchi and Beaufort Seas, as well as in the Hudson and Baffin Bays.

The LAON assimilation of SIC also improves the SST simulation along the sea ice edge throughout the year, with the most pronounced areas in the Greenland and Barents Seas. However, the assimilation generally has little impact on the ice-free area in the North Atlantic Ocean.

— The LAON assimilation generally has a minor impact on the simulated SSS. The current NorHAPS reproduces the SSS very well in the ice-free North Atlantic Ocean. However, it tends to produce a large SSS bias in the Arctic shelf seas, which

highly likely results from inaccurate river discharges, precipitation and evaporation in the model. A further investigation is needed to mitigate this deficiency.

LAON is an efficient and accurate data assimilation method. It also has a high capability in simulating the sub-daily evolution when only daily observation is available. These advantages provide a very promising basis for further application in global high-resolution coupled models. In the present study, due to the large bias in the SIT field, our focus has been mainly on the

evaluation of the model analysis. Further evaluation of model predictions will be performed in a following study with additional SIT assimilation.



*Author contributions.* KW developed and implemented the LAON method in CICE. KW and AA prepared the data and performed the model simulations. KW, AA and CW analyzed the results. KW wrote the draft manuscript, and all authors discussed the results and refined the manuscript.

*Competing interests.* The authors declare that they have no conflict of interest.

*Code availability.* The coupled HYCOM-CICE model is available through https://github.com/nansencenter/NERSC-HYCOM-CICE. The source code of LAON for assimilating the SIC in the coupled HYCOM-CICE model is available through https://doi.org/10.5281/zenodo.7572286 tagged as version 0.1.1 of the hycom-cice_coin repository.

*Data availability.* The model twin experiment data is available at https://doi.org/10.5281/zenodo.7533372. The AMSR2 SIC is available at
https://seaice.uni-bremen.de/data/amsr2/. The SSMIS SIC is available at ftp://osisaf.met.no/archive/ice/conc. The NIS ice chart is available at https://doi.org/10.48670/moi-00128. The USNIC ice chart is available at https://usicecenter.gov/Products. The NEMO SIC is available at https://doi.org/10.48670/moi-00016. The TOPAZ4 SIC is available at https://doi.org/10.48670/moi-00001. The neXtSIM SIC is available at https://doi.org/10.48670/moi-00004. The weekly mean CS2SMOS SIT is available at ftp://smos-diss.eo.esa.int/SMOS/. The OSTIA skin SST is available at https://doi.org/10.48670/moi-00167. The ISAS SSS is available at https://doi.org/10.48670/moi-00037.

*Acknowledgements.* This study was supported by the Norwegian Research Council project 4SICE (grant No. 328886), the Norwegian FRAM Flagship program project SUDARCO (grant No. 551323), the Nordic Research Council project NOCOS DT, and the National Key R&D Program of China (grant No. 2022YFE0106300). The authors are grateful to Gunnar Spreen in the University of Bremen for the discussion about the AMSR2 SIC, and Nick Hughes in the Norwegian Ice Service for the discussion about the ice charting.



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
