# Peer review of "A Local Analytical Optimal Nudging for assimilating AMSR2 sea ice concentration in a high-resolution pan-Arctic coupled ocean (HYCOM 2.2.98) and sea ice (CICE 5.1.2) model"

_The Cryosphere, 2023_

## Author Comment (AC1)

**Reply to reviewer 1**

**1 Minor comments**

(S=Section, P=Page, l=Line)

• *Abstract : The authors call the general protocol as a "twin experiment". It is not a classical way to define realistic data assimilation experiments using real data. In the scientific literature, the expression "twin experiment" defines, generally, data assimilation experiment based on simulated observations. This is not the case here, and I think, like that, it is confusing for the readers.*

*I strongly suggest to modify this presentation of the experiment protocol.*

Answer: Thanks for pointing this out. We have changed "twin experiment" to "hindcast experiment".

• *Abstract : "The SIC innovation (model minus observation)". In data assimilation paper, the innovation is defined as the opposite (Observation minus model). In addition, after in the equations, the author present the correct expression or formulation. I suggest to correct this sentence.*

Answer: Thanks. It is corrected.

• *S.2.2, l.200 : Please, define hbar.*

Answer: We have added hbar before the equation, which denotes the estimated actual SIT.

• *S.3.1.1, l.224 : "...being about 0.25 when SIC = 0 ...". I am agree for oceanic areas which are very close to the marginal zone, but I think the uncertainty should be more precise for location which are very far from the sea ice extent. The authors should say something about that.*

Answer: we have added the following short discussion after the sentence:

It is noted that the uncertainties calculated here are fully based on the AMSR2 radiometric properties, the tie-point variability and the atmospheric opacity. For the open water and MIZ close to the sea ice edge, the high uncertainty is generally realistic and implies that the observed ice edge may be not very accurate. However, for the ice-free areas far away from the sea ice edge (Fig. 3b), the uncertainty should be much lower due mainly to the much higher SST. Such impact is not considered in the present study. Nevertheless, the high uncertainty in these ice-free areas

should generally be of little effect on the assimilation of the sea ice cover, as the warm ocean surface would enhance the maintenance of the ice-free situation.

- *S.4, l.324 : "A twin experiment…" Same discussion as before*

Answer: corrected.

- *S.4, l. 327-332 : The discussion about the Anomaly Correlation Coefficient is not necessary. The authors explain obvious things and basic statistics with an example. In addition, this metrics is not showed in the paper so why do the authors talk about it? I suggest to remove these comments.*

Answer: removed.

- *S.7, l. 573: "A twin experiment…" Same discussion as before*

 Answer: corrected.

**2 Other Comments**

(S=Section, P=Page, l=Line)

- *S.2.1, l.83 : "parmaterized" should be "parameterized"*

Answer: corrected.

- *S.4, l.324 : "squre" should be "square"*

Answer: corrected.

- *S.4, l.373 : "the" is written twice*

Answer: removed one "the".

- *Reference : the reference for Wang should be timely ordered.*

Answer: The references here follow the TC template in Latex. The first Wang has only 2 authors, and the second has more than 3 authors, so the order does not follow the time.

Others: we have also added the units for Figures 6, 8 and 9.

---

## Author Comment (AC2)

Response to Reviewer 2

*1. The reason given here that there is some risk of misleading conclusions from the use of ACC makes sense, but I do not think it is a sufficient reason to discard the ACC metric. Perhaps the authors could consider a newly developed metric that integrates ACC and RMSE (Hu et al., 2022). Obviously, changing the metrics in this way does not change the conclusions of this paper, so if the authors insist on using RMSE, I would suggest deleting the "reasons for not using ACC" due to its inadequacy and misleading.*

Answer: Thanks for the information. We have removed the ACC discussion here.

2. There are several issues with the figures:

*3. As many geographic locations are mentioned in the text, I recommend adding labels to the map provided in Fig. 1 to enhance the readability, including marking the two points in Fram Strait and Nansen Basin.*

Answer: Figure 1 is replaced by a new figure with ocean and sea names together with bathymetry for the model domain. It is now clear to see the second point should be in the northern Barents Shelf, rather than Nansen Basin originally. Thanks for the comment.

*4. The meaning of should be clarified in the legend of Fig. 2.*

Answer: We have added explanations for the legend as follows: Here OI, 3DVar, LAON and EnKF represent optimal interpolation, three-dimensional variational, local analytical optimal nudging, and ensemble Kalman filter, respectively. In the lower legend, the EnKF mean is an average of all the EnKF members.

*5. I found it is quite confusing that the all the color bars in this paper extend at both ends. The SIC should not have values greater than 1 or less than 0, and standard deviation and SIT should not have values less than 0, right? Please modify the color bars in Fig. 3, Fig. 4, Fig. 6, Fig. 9(i-l), and Fig. 10.*

Answer: Thanks for pointing this out. We have made corrections for the extend.

*6. If we assume that the monthly averaged SIC field for March 2022 in Fig. 4 is an approximate representation of March 16 2022, then it seems safe to conclude that NorHAPS Free_run's*

*simulation of the MIZ is already better than TOPAZ4? I believe it would be interesting to present the results of Free_run in Fig. 10 and give its IIEE and IME time series in Fig. 11 as well, and this will deepen our understanding of the benefits of using LAON to assimilate SIC.*

Answer: We have added the NorHAPS Free_run in Fig. 10, and added IIEE and IME time series in Fig. 11. Their statistical properties are also added in Table 1.

For describing Fig. 11, we have added the following paragraph: As has been shown in Fig. 5, the Free_run generally has a large bias during the summer period. However, the bias is mainly located in the central Arctic and Pacific side marginal seas (Fig. 4), particularly in Beaufort, Chukchi and New Siberian Seas. For the European Arctic as shown by the NIS ice chart (Fig. 10), the bias is generally moderate (see Fig. 4). This can also be seen in the IIEE and IME (Fig. 11). Except for the summer IIEE which is apparently higher than the other products (Fig. 11a), the Free_run IIEE and IME are generally comparable to the largest IIEE and IME of the other products, although significantly higher than those of the LAON.

For describing the table, we have added the following sentences: The NorHAPS Free_run has the highest mean IIEE and IME, and the NorHAPS LAON has the lowest IIEE and IME among all the products (Table 1). On average, the IIEE and IME of the Free_run are about 70% and 55% higher than those of the LAON.

*7. In Fig. 12, another thing that impressed me about NorHAPS LAON is the underestimation of the high SIC in the MIZ (e.g., 8-15 April 2021), which captured by neXtSIM. I think it is necessary to mention this in the text, and if possible, it would be better to give a one-sentence comment.*

Answer: we have added the following descriptions: However, for the April 2021 case (Fig. 12a), while the NorHAPS LAON generally captured the evolution of the SIC, it tends to underestimate the high SIC, particularly during 7-10 April 2021. By contrast, this process is very well captured by neXtSIM. A further improvement of the NorHAPS model and assimilation system is therefore highly desirable.

*8. Maybe I misunderstand something, but the ice melting and brine rejection (related to ice freezing) have been identified as playing a dominant role in sea surface salt fluxes (e.g., Lambert et al., 2018). Except for the open water along the coast which is directly influenced by river runoff, I might expect SSS in the Arctic to benefit from SIC assimilation as the SIC spatial distribution is much more realistic. Although the authors argue that this large SSS bias may come from salinity drift generated by the 10-year spin-up run, I would suggest at least mentioning why the optimization of the SIC did not improve the SSS in the sea ice covered area.*

Answer: Thanks for the comment. We have replotted the figure and added the SSS difference between the LAON and Free_run in Figure 9. One paragraph is also added to describe the effect of SIC on the SSS, as follows:

The SSS changes due to the SIC assimilation are much weaker than the absolute SSS bias in the present study. For better illustration, the monthly mean difference between the LAON SSS and Free_run SSS is calculated (i-l in Fig. 9). While generally weaker than the absolute bias, the absolute difference over 1 PSU can often be seen in a remarkable part of the Arctic Ocean, particularly in the shelf seas. During the melting season (Fig. 9j), similar to the effect on the SST (Fig. 8), the SIC assimilation notably increases the SSS along the coasts of Beaufort, Chukchi, New Siberian, Laptev and Kara Seas. The assimilation removes the overestimated sea ice there, thus mitigating the SSS decrease due to the unrealistic sea ice melting. During the freezing period (k, l in Fig. 9), unlike the situation for SST which would generally remain close to the freezing point, SSS tends to continuously increase due to the sea ice freezing and brine rejection. The SSS difference between the LAON and Free_run simulations here is most probably caused by the different sea ice freezing speeds, which are mainly controlled by the corresponding SIT and snow depth. The relatively lower SIT in the LAON simulation (Fig. 6) would foster a more rapid freezing and therefore larger SSS increase during the freezing period. Such a positive increment (Fig. 9l) is seen to counterpart the negative bias in the Arctic (Fig. 9d), thus improving the SSS simulation. This is consistent with the findings by Lambert et al. (2019), who identified that sea ice melting and brine rejection due to sea ice freezing play a dominant role in the Arctic SSS flux in their perfect model experiment using a strong SSS climate restoration. It remains to see whether longer time SIC assimilation can further improve the SSS simulation when no SSS climate restoration is applied.

 We have also changed the conclusion on the effect of SIC assimilation on SSS, as follows:

The current NorHAPS reproduces the SSS very well in the ice-free North Atlantic Ocean. However, it tends to produce a large SSS bias in the Arctic, particularly in the shelf seas, which highly likely results from inaccurate river discharges, precipitation and evaporation in the model, as well as possible inaccuracy in the initial condition in the Arctic. The SIC assimilation generally has a weaker effect on the simulated SSS than the model SSS bias. Nevertheless, the LAON SIC assimilation provides a reasonable description of the seasonal SSS response due to the optimization of the sea ice cover. During the melting season, the SIC assimilation removes the overestimated sea ice, thus mitigating the SSS decrease due to the unrealistic sea ice melting. During the freezing period, the SSS tends to continuously increase due to the sea ice freezing and brine rejection resulting from the relatively lower SIT. A further investigation is needed to mitigate the large model SSS bias.

Other issues:

Line 11: As "SST" is not used again in the abstract, the abbreviation should be deleted here.

Answer: It is true "SST" is not used again in the abstract, but for indexing and searching purposes of the abstract, we feel having "SST" here is useful, so it is not removed.

*Line 22: Sumata et al. (2023) may be another relevant work worth citing.*

Answer: Thanks. We have added this reference.

*Line 46: May be "The original empirical treatments … have been upgraded …"*

Answer: Corrected.

*Line 62: Since this is the first time you use the abbreviation "NorHAPS" in the main text, it should be spelled out here.*

Answer: Added. The sentence now turns to: The Norwegian High-resolution pan-Arctic ocean and sea ice Prediction System (NorHAPS) is a developing modeling and prediction system at the Norwegian Meteorological Institute.

*Line 75: "bathmetry" should be "bathymetry"*

Answer: corrected.

*Line 80: "These" should be "This"*

Answer: corrected.

Line 81: "parmaterized" should be "parameterized"

Answer: corrected.

*Line 90: Maybe remove "the". Additionally, I strongly recommend that the authors conduct a detailed grammar check, especially for the application of prepositions; there are too many improper uses of such in this paper to list here.*

Answer: With the reference here, it should be fine with the "the".

A grammar check has been performed to improve the prepositions.

*Line 104: "five category" should be "five-category"*

Answer: corrected.

*Line 106: "account" should be "accounts"*

Answer: corrected.

*Line 124: "towards" should be "toward"*

Answer: corrected.

*Line 138: "avoid" should be "avoids"*

Answer: corrected.

*Line 215: Add spaces after the comma and "version"*

Answer: added.

*Line 248: "These satellites data" should be "These satellite data"*

Answer: corrected.

*Line 259: Maybe remove "for"*

Answer: removed.

*Line 279: grid spacing "is" approximately …*

Answer: No"is" looks better.

*Line 286: "a hourly" should be "an hourly"*

Answer: corrected.

*Line 300: "Institut" should be " Institute"*

Answer: corrected.

*Line 313: "Institut" should be " Institute"*

Answer: not found.

*Line 318: "access" should be "accessed"*

Answer: corrected.

*Line 327: "Squre" should be "Square"*

Answer: whole sentence removed.

*Line 330: "judgement" should be "judgment"*

Answer: whole sentence removed.

*Line 332: "disrepancy" should be "discrepancy"*

Answer: whole sentence removed.

*Line 373: Two consecutive "the"*

Answer: removed one "the".

*Line 378: "The MYI are" should be "The MYI is"*

Answer: corrected.

*Lines 393-395: "A detailed check of the sea ice velocity field …", this is an interesting point, but unfortunately I could not find a figure illustrating this.*

Answer: This sentence now changed to: A detailed check of the sea ice velocity field (not shown) indicates that ... . We will study this problem with the assimilation of sea ice thickness later.

*Lines 416-417: "It is noteworthy that these large bias areas are generally collocated with the larger uncertainty areas in the observed ISAS SSS (i−l in Fig. 9). " This is not a persuasive description, the four patterns of ISAS STD look quite same for me, but there is a clear difference in the model (Fig. 9e-h), the large bias areas in the model does not seem to correspond well with the large STD region in ISAS.*

Answer: We have changed the sentence to limit the description for the central Arctic and use "often" to replace "generally", as follows: It is noteworthy that these large bias areas in the central Arctic are often collocated with the large uncertainty areas in the observed ISAS SSS (e−h in Fig. 9).

*Line 419: This would induce "an" overestimate …*

Answer: corrected.

*Line 471: of "the" largest …*

Answer: corrected.

*Line 500: "produces" should be "produce"*

Answer: corrected.

*Line 549: has been reached "on" whether …*

Answer: corrected.

*Lines 597: "LAON is an efficient and accurate data assimilation method." While you have adequately described the accuracy aspects in the conclusion, I believe your readers would benefit greatly from a more detailed explanation of the term "efficient". For example, add some words like "The extra computational cost for the LAON assimilation is negligibly small, being about 5% of the free run in the present study. " (Lines 43-44) here.*

Answer: Thanks for the comment. We have added the description after the sentence. We also added it in the abstract.

**All the doi numbers have been added to the references:**

*Line 656: Maybe provide the url link: [http://library.arcticportal.org/id/eprint/1671](http://library.arcticportal.org/id/eprint/1671)*

*Line 657: The doi number is:*

*10.1175/1520-0442(2003)016<0571:BPOASF>2.0.CO;2*

*Line 675: The doi number is:*

*10.1175/1520-0485(1979)009<0815:ADTSIM>2.0.CO;2*

*Line 676: The doi number is:*

*10.1175/1520-0493(1980)108<1943:MAVTSI>2.0.CO;2*

*Line 677: The doi number is:*

*10.1175/1520-0485(1997)027<1849:AEVPMF>2.0.CO;2*

*Line 703: The doi number is: 10.1175/1520-0493(2004)132<1341:MSITUI>2.0.CO;2*

*Line 708: The doi number is: 10.1002/qj.49711247414*

*Line 711: The doi number is: 10.1007/BF00211684*

*Line 716: The doi number is: 10.1029/JC092iC07p07032*

*Line 741: The doi number is: 10.1080/01431169608949096*

*Line 749: The doi number is:*

*10.1175/1520-0493(1990)118<1250:UOFDDA>2.0.CO;2*

*Line 725: I suppose it should be "Arctic Shipping Status Report"*

Answer: You are right. We have changed the title to: "Arctic Shipping Status Report (ASSR) #1: The Increase in Arctic Shipping 2013-2019"

---

## Author Response (AR1)

**Response to Editor's comment**

*It is great to see improvements for sea ice volume when you assimilate sea ice concentration. I wonder whether the choice of sea ice concentration product has an impact on the sea ice volume changes? Can you give some reasons why thickness errors remain high in certain areas and improve more in other areas. A short discussion about this would be enlightening.*

**Reply**: Thank you very much for pointing this out. We have added two paragraph to discuss the effect of SIC assimilation on the SIT and SIV at the end of the section, as follows:

When combining the seasonal evolution of the SIC and SIT fields (Figs. 4 and 6), we can see that the effect of the SIC assimilation on the SIT is the most significant during the summer period. The LAON assimilation effectively corrected the large bias in the spatial distribution of the SIC, particularly in the Arctic shelf seas. Such a correction not only rectifies the summer SIT bias in these areas, but also provides an open ocean condition close to the observations for the later new ice development during the freezing period. By contrast, the persistent large overestimate in the MYI suggests that the SIC assimilation tends to have limited SIT improvement for the ice that survives through the melt season. Such a mechanism is unanimously applicable to the LAON SIC assimilation when using other SIC products, although the improvement may differ due to the variations in the SIC values and uncertainties.

The effect of the SIC assimilation on the SIT implies that the LAON SIC assimilation would also improve the SIV using any reasonable SIC products, with the largest improvements in the melt season and from new seasonal sea ice formed during the freezing period. While the improvement in the surviving MYI is generally limited, such MYI is expected to transport out of the Arctic in several years. It is therefore anticipated that, after several years' SIC assimilation, the SIT spatial distribution and the SIV would have an overall improvement. Nevertheless, a direct SIT assimilation would be more effective and prompt.

We also changed the corresponding conclusion as follows,

The LAON assimilation of SIC improves the simulation of SIT and SIV, with the largest improvements in the melt season and from new seasonal sea ice formed during the freezing period. In the present study, the spatial pattern of the simulated SIT is noticeably improved after a one-year LAON assimilation. The LAON assimilation also reduces the overestimate of SIV, with the bias in the second year being less than 3000 km$^3$ compared with over 4000 km$^3$ in the Free_run simulation (Fig. 7). However, the LAON assimilation of SIC generally has limited improvement in the surviving MYI, and it may take several years of assimilation to reach a notable improvement. Therefore, a direct assimilation of the SIT is highly needed.

(In addition to the above paragraph, Fig. 7a is updated in which the previous SIV calculation missed to multiply the grid area. The volumes in the text have also been corrected.)